# OncoRisk: a state-of-the-art web server for bridging the oncogenic databases and pan-cancer cohorts to the translational oncology
Xiya Song [1,6], Emre Green[1,6], Xinmeng Liao[1], Hasan Turkez[2], Gozde Yesil [3], Bayram Yuksel [3], Mathias Uhlen [1], Cheng Zhang [1,4] & Adil Mardinoglu [1,5] ✉

Accurate interpretation of genomic variants remains a major bottleneck in precision oncology, due in part to fragmented knowledge across databases and limited integration between clinical evidence and population-scale genomic datasets. Here we present OncoRisk, a stand-alone, user-friendly web server that unifies data from over ten oncogenic databases and seven large-scale pan-cancer cohorts, enabling rapid multi-database queries and network-based exploration of genomic variants, gene-gene interactions, and therapy associations. The platform features a semi-automated reporting workflow that generates comprehensive, patient-specific clinical reports from raw tissue sequencing data and categorizes variants into actionable tiers. For translational research, OncoRisk provides modules for data-driven exploration, allowing users to validate findings by interrogating mutation frequencies and clinical associations across real-world patient data. Furthermore, an integrated suite of analytical tools enables comprehensive, cohort-level investigations of mutational landscapes, prognostic biomarkers, and oncogenic signaling pathways. By providing a unified ecosystem that bridges curated knowledge with large-scale cohort data, OncoRisk serves as an effective catalyst for both discovery research and clinical application in oncology. OncoRisk is publicly available at https://www.phenomeportal.org/oncorisk.

Genomic variations, including point mutations, gene fusions[1], structural variations, and copy number variations[2], act as fundamental drivers of cancer development[3]. Driver mutations lead to tumorigenesis by altering protein functions, dysregulating gene expression levels, and perturbing metabolic, regulatory and signaling pathways[4]. Genomic profiling has markedly expanded the range of clinical diagnostics and therapeutic options for cancer patients[5]. To date, over 100 FDA-approved mutation-specific therapeutics have entered clinical practice, providing actionable biomarkers that inform targeted therapeutic decisions cancer therapy[6].

The variant allele frequencies (VAFs)[7], along with the functional impacts of each mutation, differ across individuals, tissues, and tumor subtypes[8]. This leads to heterogeneous mutational landscapes, which are crucial for tumor classification, prognosis prediction, and personalized medicine. Thus, the advance of precision oncology relies on accurately

identifying, interpreting genomic mutations, and elucidating their clinical implications. Community-curated resources, such as ClinVar (version 2025)[9], CiVIC[10], and OncoKB[11,12], provide variant- and gene-level clinical interpretations with therapy information. Artificial intelligence (AI) empowered algorithms, such as CancerVar[13], have further enhanced clinical interpretations following the AMP/ASCO guidelines[14]. Despite their utility, existing platforms remain isolated, lacking efficient cross-database search functionality and integrative analytical capabilities. Navigating the "sea of oncogenic evidence" remains a laborious and time-consuming process for research and real-world clinical translation, as the numerous variants identified through tumor sequencing require parallel investigations across multiple knowledge bases.

On the other side, large-scale pan-cancer cohorts, represented by the Cancer Genome Atlas (TCGA)[15], have driven deep molecular profiling of

[1]Science for Life Laboratory, KTH Royal Institute of Technology, Stockholm, Sweden. [2]Department of Medical Biology, Faculty of Medicine, Atatürk University, Erzurum, Turkey. [3]Phenome Omics R&D, Mehmet Ali Aydinlar Acibadem University, Istanbul, Turkey. [4]The Roger Williams Institute of Liver Studies, Faculty of Life Sciences & Medicine, King's College London, London, UK. [5]Centre for Host-Microbiome Interactions, Faculty of Dentistry, Oral & Craniofacial Sciences, King's College London, London, UK. [6]These authors contributed equally: Xiya Song, Emre Green. ✉e-mail: adilm@scilifelab.se; adil.mardinoglu@kcl.ac.uk

patients across diverse tumors. The cutting-edge advances in the field are emerging pan-cancer projects other than TCGA, such as pan-cancer analysis of whole genomes (PCAWG)[16], MSK-CHORD[17], China pan-cancer cohort[18], and AACR Project GENIE[19] dataset, which comprises over 211,000 sequenced cancer patients. However, findings from these studies are often analyzed and published in isolation, with limited accessibility for code-free querying and cross-cohort comparison. Directly applying existing mutation frequency data to support variant interpretation and validation in research or clinical practice remains uncommon and challenging. Manually bridging this gap involves cross-referencing across multiple websites and platforms, which leads to inefficiency. Cohort exploration tools, represented by cBioPortal[20,21], offer unified access to cross-project-level cancer genomics data. Despite this utility, they are not designed for variant-centric lookup and for comprehensive implementation of mutation analysis and visualizations, such as generating oncoplots, characterizing mutation signatures, or assessing variant-based survival associations. Currently, researchers and clinicians face challenges in accessing comprehensive, cross-linked information, including allele and mutation frequencies, structural and pathway-level insights, and validation across real-world cancer cohorts. A unified system is needed to integrate this knowledge, enabling researchers to

accelerate translational studies and clinicians to directly apply these insights to individual patients.

Motivated by this need, we developed OncoRisk (https://www.phenomeportal.org/oncorisk), a user-friendly, stand-alone web server that unifies knowledge-based resources, real-world cohort data, and clinical and therapeutic information. OncoRisk enables single or batch queries of variants, genes, and other oncogenic terms across multiple databases and reveals their interconnections. It also enables the end-to-end generation of comprehensive, patient-specific oncology reports from somatic mutation data in VCF format. Moreover, OncoRisk facilitates translational research by enabling rapid queries of genes or variants in pan-cancer cohorts, as well as the analysis of cancer cohorts through a series of tool sets. We envisage OncoRisk as a robust bridge linking biological discoveries with their translational and clinical implementation.

## Results

### Overview of OncoRisk's structure and functionalities

OncoRisk integrates more than ten databases and builds its overall architecture upon four interconnected pillars (Fig. 1): (1) it aggregates major precision oncology knowledge bases (POKBs), such as CiVIC and OncoKB,

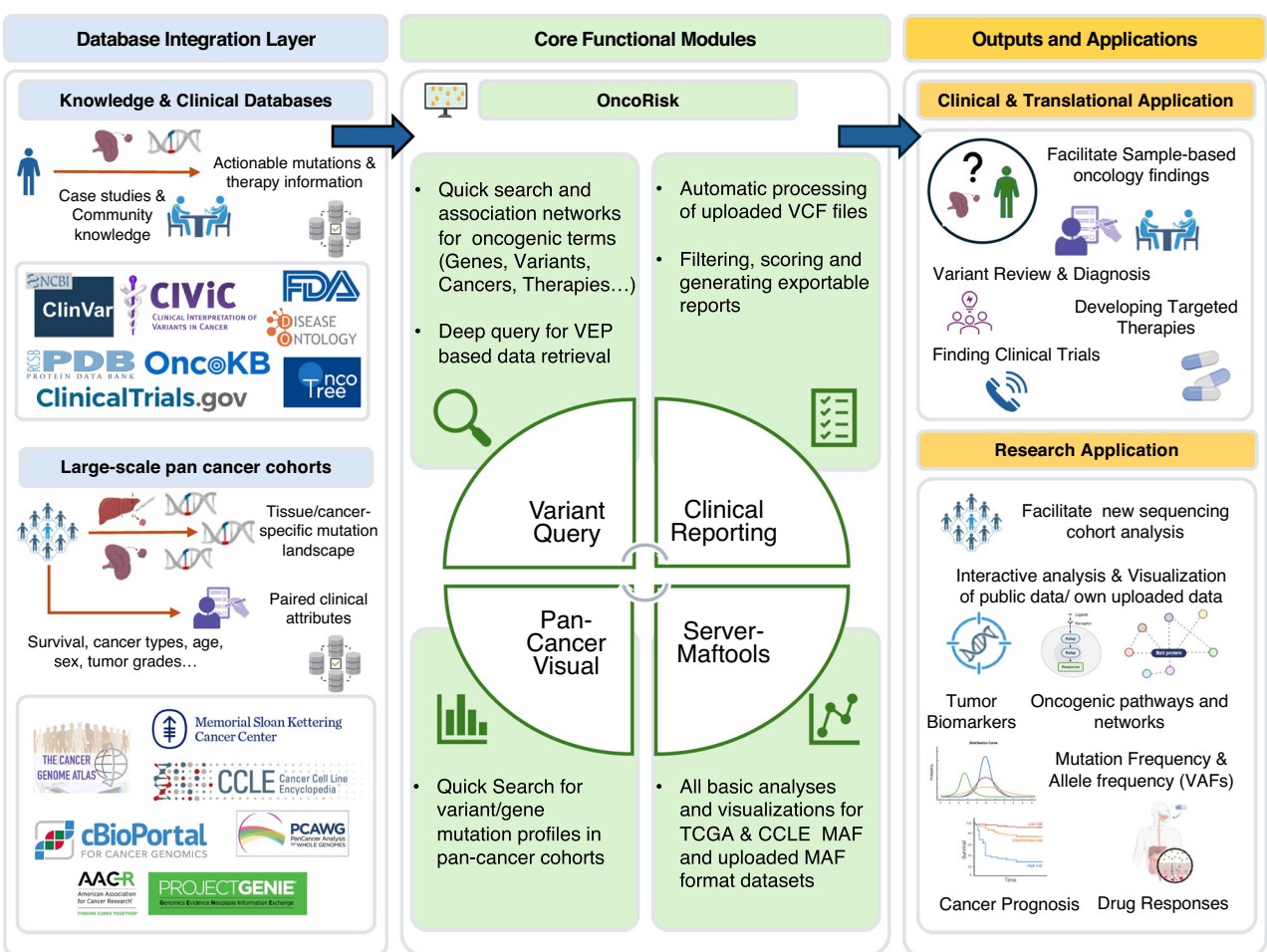

**Fig. 1 | Overview of the OncoRisk web server.** From the database integration layer (light blue panel), OncoRisk integrates more than ten knowledge-based and clinical databases, as well as large-scale pan-cancer cohorts, which serve as the foundation for building its functional modules. The four primary functional modules are enclosed in light green rectangles: variant query (including knowledge base exploration, quick search, network tools, and deep query), pipeline analysis for clinical reporting, pan cancer visualization (pan-cancer-visual), and analysis module for MAF cohort files (server-maftools). Together, these modules provide unified access to genomic knowledge bases, real-world cohorts, and user-uploaded data, supporting rapid variant and gene queries, automated or customizable patient-level

reporting, validation of mutation frequencies across pan-cancer cohorts, and comprehensive cohort-level visualization and analysis. The outputs and applications benefit two major aspects, as indicated in the orange panel. For clinical and translational applications, OncoRisk provides strong technical support for clinicians by enabling the fast generation of reports for reviewed mutations, supporting patients, accelerating the development of targeted therapies, and facilitating patient allocation to potential clinical trials. For research, OncoRisk supports interactive analysis of sequencing cohorts, allowing identification of tumor biomarkers and oncogenic pathways, comparison of mutation frequencies, and prediction of cancer prognosis and drug responses. Figure 1 contains elements created with BioRender.com.

to support rapid variant queries and the exploration of association networks among oncogenic terms, while also introducing unique features for deep queries, including cancer family trees, terminology association networks representing direct and indirect relationships, and protein-mutant-drug 3D visualizations. The queried results from quick search and deep query can be exported via downloadable JSON outputs. (2) Based on the single variant's deep query function, OncoRisk implements a semi-automated workflow for patient-specific somatic mutation analysis, generating diagnostic, prognostic, and therapeutic reports through a dedicated interface that allows manual adjustment and exportation of organized outputs (JSON and PDF), thereby accelerating the process from identifying actionable variants to delivering information back to patients. (3) The "pan-cancer-visual" module enables variant-based and gene-based queries across seven pan-cancer cohorts, providing rapid validation of variants or genes of interest by assessing their mutation frequencies in real-world cancer cohorts and facilitating the reuse of large existing datasets. (4) The "server-maftools" module supports efficient cohort-level mutation visualization and analysis, enabling the exploration and validation of novel biological insights, such as prognosis-associated mutations and enriched oncogenic pathways. It provides mutation data from 34 TCGA cohorts and CCLE cell lines as ready-to-use examples, while also accepting any of the user-uploaded data for custom analysis.

## Quick-query interface and network tools for multi-query visualization

The OncoRisk knowledge Base was designed and built based on CiVIC for its fully open-source, highly structured, and massive community-based knowledge records. Then, other databases (OncoKB, ClinVar, and CancerVar) were complemented to provide comprehensive integrated information. First, we selected and retrieved 9 oncogenic term categories from the CiVIC public API and cached them in local storage. These terms were categorized as 169 clinical assertions (a collection of evidence Items), 11,148 clinical evidence items, 4016 molecular profiles (complex combinations of one or more CIViC variants), 899 features (genes and fusions), 3890 variants, 435 disease items, 578 therapies, 272 phenotypes (symptoms or abnormalities) and 3987 sources (reference of evidence items) (Fig. 2A). All the items are shown from the side panel of the OncoRisk homepage. Information is organized as follows: (1) clicking a category reveals its full list of items, and (2) selecting an item displays detailed information. Then, from the detail page, information from ClinVar and OncoKB is retrieved for variants or gene fusions and integrated into the corresponding molecular profiles and features page.

Using this as a basis, we developed two ways, "quick search" and "network tools", to retrieve information from the knowledge base. Both support flexible synonymous search, allowing the use of multiple genomic identifiers and nomenclatures to obtain relevant information.

The first query function, "quick search" box, is designed for fuzzy matching any of a single oncogenic term of interest across all term categories, including gene names, fusions, or variants formatted with amino acid alteration (Fig. 2B). This function also supports searches by variant nomenclature, including rsID; for example, querying "rs1134880" returns the *BRAF* V600E profile, and vice versa, searching *BRAF* V600E retrieves its rsID within the corresponding detail page. Except single variant, the query, such as "*BCR::ABL1*", returns ten molecular profiles corresponding to this specific gene fusion, and a query, such as "lung cancer", returns the specific disease-focused page. At the bottom of the query box, users can adjust which category of oncogenic terms should be included in the fuzzy search results.

Second, OncoRisk provides the "network tools" tab, which visualizes single- or multi-term queries interactively and supports simultaneous querying of two or more terms (Fig. 2C). Users can submit queries directly from this tab. The network displays the queried items, together with their interconnected nodes across different ontological categories, distinguished by colors. Disease terms are grouped by tissue (yellow) using information retrieved from DO and OncoTree. All relevant terms matching the search query are displayed at the top of the network page. Clicking any of these terms centers the view on the corresponding node and zooms in for detailed

inspection. On the bottom right, users can toggle nodes on or off based on term categories. As an example, we showed the association networks generated by searching for *EGFR* and erlotinib together. (1) The network shows that erlotinib was strongly connected to *EGFR* through a total of 666 evidence records (orange) from CiVIC, such as EID4220, which further search by EID4220 and *EGFR* L747P indicates that the *EGFR* L747P mutation is associated with a poor response to erlotinib (Supplementary Fig. 1). (2) Adding L858R, the most common and well-known *EGFR* point mutation, into the network retrieved two CiVIC assertion records (red), AID105 and AID5. These assertions showed that L858R is one of the most common sensitizing *EGFR* mutations in NSCLC, for which tyrosine kinase inhibitors, such as gefitinib and erlotinib, improve progression-free survival compared to chemotherapy in patients carrying this mutation (Fig. 2C). (3) Gain-of-function and Level 1 evidence from OncoKB, together with Tier 1-strong attribute from ClinVar, is displayed from the associated molecular profiles "*EGFR* L858R OR *EGFR* exon 19 deletion (Fig. 2D and Supplementary Fig. 2)." (4) The network also presented over 20 additional drugs (green) associated with L858R mutations, including afatinib and gefitinib. Interestingly, the network reveals connections beyond the original query intent, such as multiple connections between T790M and L858R mutations, and shows that both L858R and the *EGFR* gene itself are directly or indirectly associated with several other tumor types in addition to lung cancer, such as high-grade glioblastoma for L858R specifically, and 34 different cancers associated with *EGFR* (Fig. 2E). This network search can also reveal the 61 therapies recorded in the database.

## Deep query interface for integrated multi-database annotation

We developed a submission-driven deep-query interface that serves as a complementary tool for analyzing any genomic variants, including those absent from CiVIC (Fig. 3A), such as novel variants identified through sequencing or variants currently lacking clinical evidence. This function presents a detailed variant-focused page for any submitted variant identifier compatible with VEP original data formats (HGVs identifiers, rsID, or one-line VCF format). In the information panels on the right side of the page, we integrated locally cached databases, annotation tools (Ensembl VEP), and real-time API access to multi-layer resources, including CancerVar, OncoKB, ClinicalTrials.gov, cancer families (OncoTree and disease ontology), and protein-drug complex 3D structures (RCSB PDB). A human anatomical model was integrated alongside the information panel to visually indicate the tissue locations of all cancer types associated with the queried variant. As a result, this interface presents detailed clinical summaries in multiple sub-tabs and gathers comprehensive data from external resources, including the following aspects: (1) the variant's basic information, including oncogenicity and pathogenicity information, along with scoring from CancerVar (Fig. 3B). A radar plot summarized this information and evidence and assigned a final tier based on the tier setting (Supplementary Data 1). Variants that contain high importance assigned by OncoKB oncogenicity or ClinVar pathogenicity, oncogenicity (ONC), and AMP tier (SCI field) will be assigned as tier 1. A detailed validation of OncoRisk tiers is attached in Supplementary Fig. 3 and Supplementary Data 2. (2) The therapeutic information from OncoKB and CiVIC, with the level of evidence (Fig. 3C). (3) Clinical trial information regarding the queried variants is provided (Fig. 3D). Using rs113488022 (the rsID for *BRAF* V600E) as a query example, a total of 68 trials are returned, of which five are currently recruiting. (4) The disease family tree, showing all cancer families associated with the queried variant, as shown as an example here for rs113488022 (Fig. 3E). As an example, papillary thyroid carcinoma is displayed using DO and the OncoTree network. Papillary thyroid carcinoma in the DO network is a subtype of papillary adenocarcinoma and a subtype of differentiated high-grade thyroid carcinoma. In OncoTree, it is retrieved as a subtype of well-differentiated thyroid cancer and eventually mapped to the thyroid tissue. While OncoTree provides no further descendants of papillary thyroid carcinoma, DO supplies more detailed subtypes (shown in purple), such as diffuse sclerosing papillary thyroid carcinoma. (5) The structure panel allows automatic retrieval of available protein complexes incorporating the

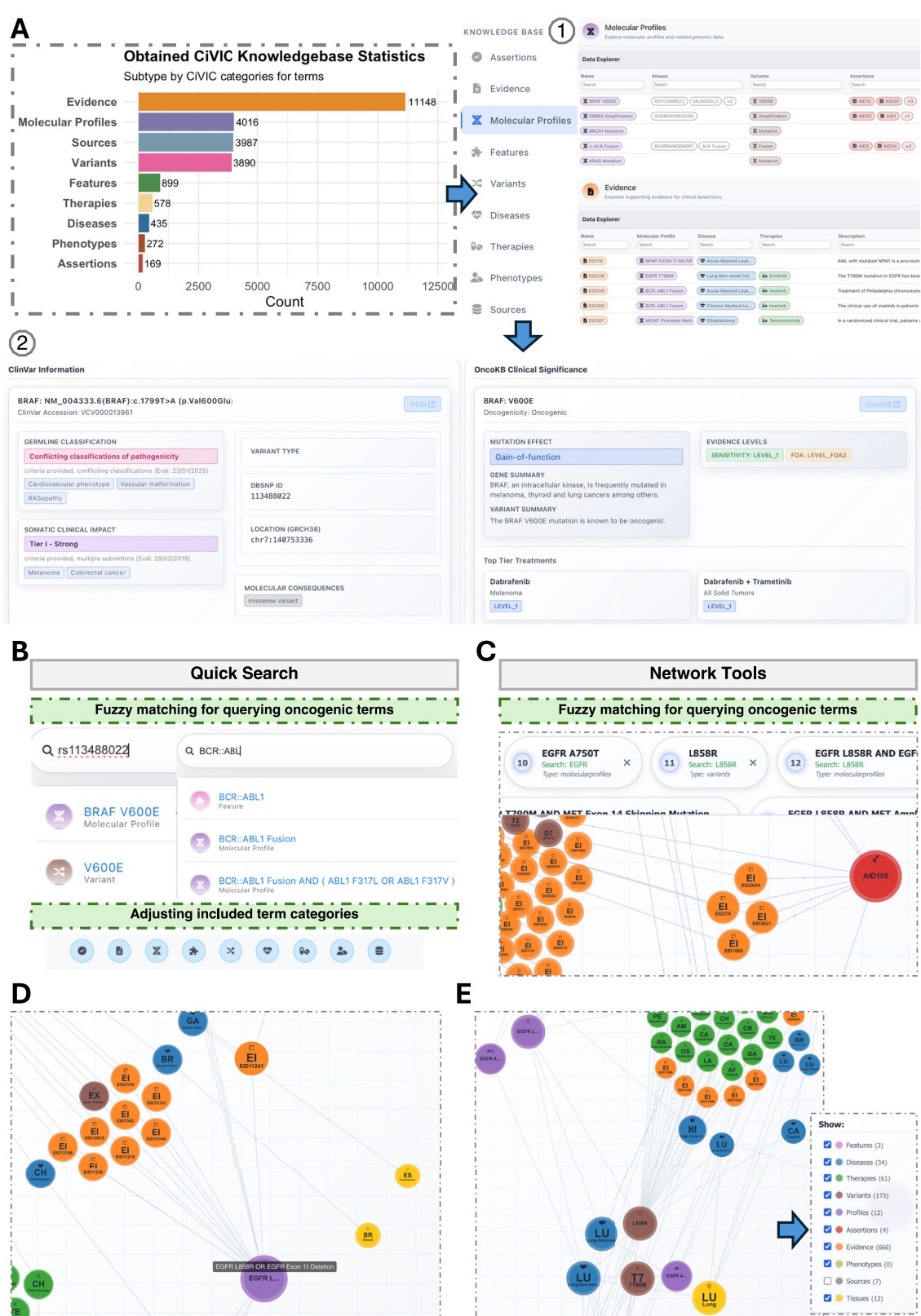

queried mutant, mutant-associated gene, and therapies with evidence (Fig. 3F). For example, trametinib complexed with *BRAF: MEK1* kinases (PDB ID: 7M0Y) is displayed. The mutant location is highlighted with red spheres, and trametinib is shown as a red chemical complex. This clearly demonstrates that trametinib targets *MEK*, not *BRAF*, and that *BRAF* V600E does not directly interact with trametinib in the 3D structural model.

**Rapid, individualized tumor profiling and reporting via an automated workflow**

In this module, we incorporated a semi-automated pipeline into OncoRisk to support comprehensive tumor profiling analysis and clinical reporting. The workflow is shown in Fig. 4A. The system enables direct upload of individual sequencing VCF files that contain thousands of somatic

**Fig. 2 | Quick-query interface, network tools, and deep-query interface for integrated multi-database annotation. A** Overview of the nine categories of oncogenic terms retrieved from CiVIC and cached locally, including clinical assertions, evidence items, molecular profiles, features, variants, diseases, therapies, phenotypes, and sources. Users can click any category to view the full list, as indicated by the numbered gray circle (1), and select specific items to access detailed information integrated with ClinVar and OncoKB, as exemplified in circle (2). **B** Quick Search interface for searching a single oncogenic term. Users can query by RS ID or gene fusions to retrieve variant-specific profiles, such as rs113488022, *BRAF* V600E, or *BCR::ABL1*. This also supports search by disease names or therapy names. Users can customize the term categories included in the search results. **C** Network

Tools for interactive visualization of single- or multi-term queries. An interactive visualization of search queries (e.g., *EGFR* and Erlotinib). Interconnected nodes represent therapeutic, prognostic, and diagnostic relationships. Red and orange nodes highlight specific CiVIC assertions (AID105, AID5) and evidence records (e.g., EID4220) linking mutations to drug responses. **D** "*EGFR* L858R OR *EGFR* exon 19 deletion" (purple node) as the associated molecular profile can be found, which provides Gain-of-function and level 1 evidence from OncoKB when searching by OncoRisk. **E** The expanded network reveals secondary associations beyond the initial query, illustrating connections between T790M and L858R mutations (brown nodes) and identifying 34 different cancer types (blue nodes), 61 different therapies (green nodes) associated with the *EGFR* gene.

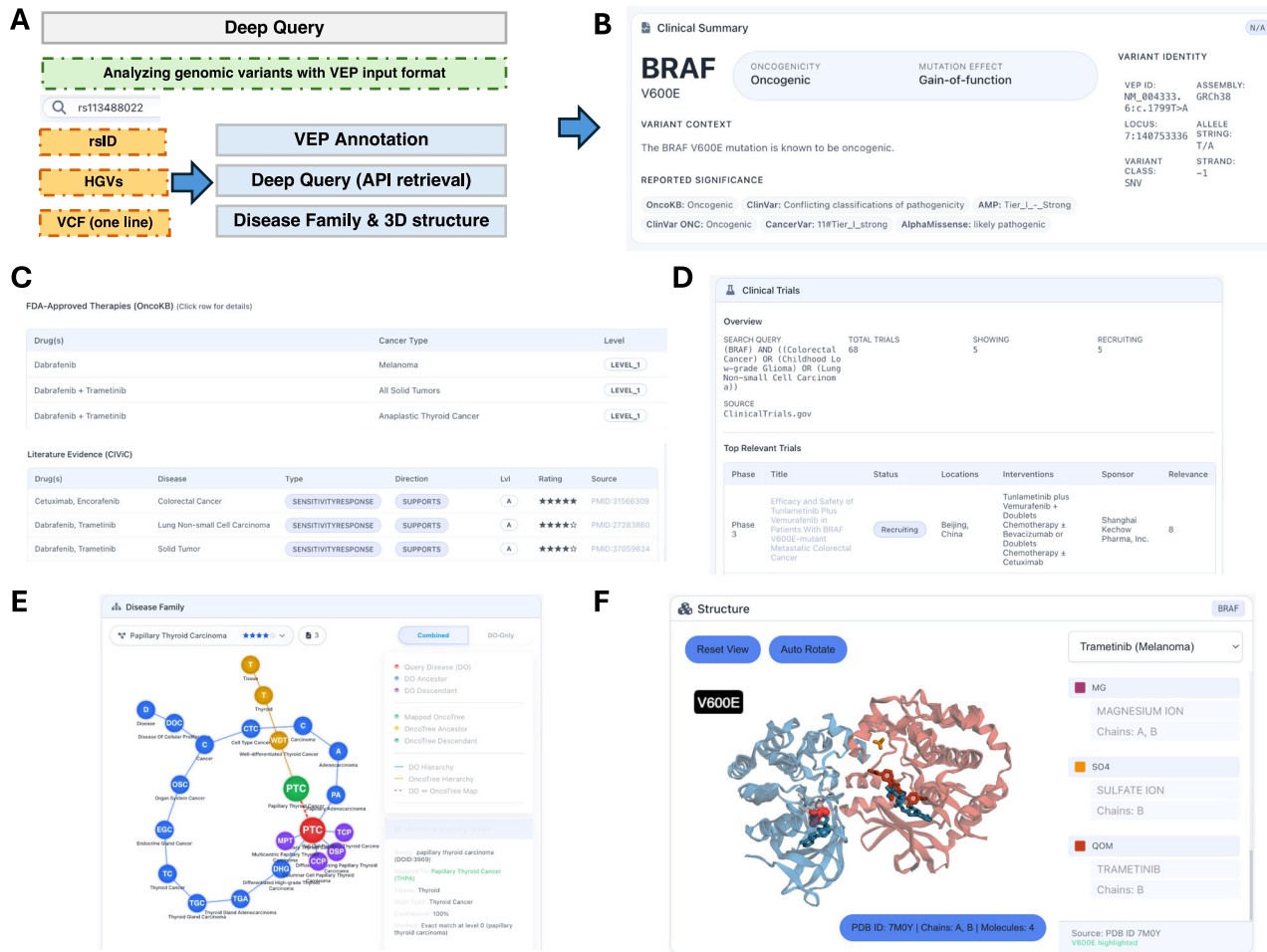

**Fig. 3 | Deep query for multi-database annotation. A** Deep query submission interface serving as the entry point for functional annotation of known or novel variants. The system accepts multiple identifiers, including HGVS, rsID, or one-line VCF formats, as indicated by the orange dashed boxes. Following submission, the platform performs VEP annotation, API-based data retrieval, and constructs disease families and 3D protein structures, as represented by the light blue solid boxes. **B** The variant's basic information, including oncogenicity and pathogenicity information, along with scoring from CancerVar. **C** Therapeutic and actionability module aggregating drug-response data and levels of evidence from OncoKB and CiVIC.

**D** Clinical trial integration provides real-time tracking of relevant trials, including recruitment status, as demonstrated by the 68 trials retrieved for the *BRAF* gene. **E** Ontological disease mapping illustrating the lineage of associated cancers through a comparative visualization of OncoTree and disease ontology (DO) networks. **F** 3D structural modeling panel showing the automatic retrieval of protein–ligand complexes, such as trametinib within the *BRAF: MEK1* complex (PDB: 7M0Y), with the mutation site highlighted to demonstrate the spatial relationship between the variant and the therapeutic agent.

mutations. The liftover (GRCh37 to GRCh38) is performed if the input file is in the GRCh37 assembly. After a quality filter to only retain variants with the "PASS" flag, these files are processed through Ensembl VEP annotation, and the initial results are presented in a structured format on the left panel of the interface. A real-time PDF preview is displayed on the right with an exportable option for gaining the original JSON file (Fig. 4B). For selected variants with CiVIC information, the deep query function is applied to provide enriched annotations from instant API access for clinical trials, OncoKB evidence, and CancerVar tiers, etc. Clinical insights are then

generated through a tier-based classification system that integrates evidence from multiple sources (see "Methods"), categorizing filtered mutations into tiers 1–4. Details of the tier assignment are shown upon clicking the "tier" information tab (Fig. 4C), where the structured JSON data (for all variants or a single variant with clinical insights) can be exported via this interface. Lastly, the pipeline outputs automated PDF reports in a user-friendly format with graphical visualization of assigned tiers, thereby enabling standardized workflows for tumor profiling that support clinical interpretation and research applications (Fig. 4D). Lastly, in the user interface, there is also a

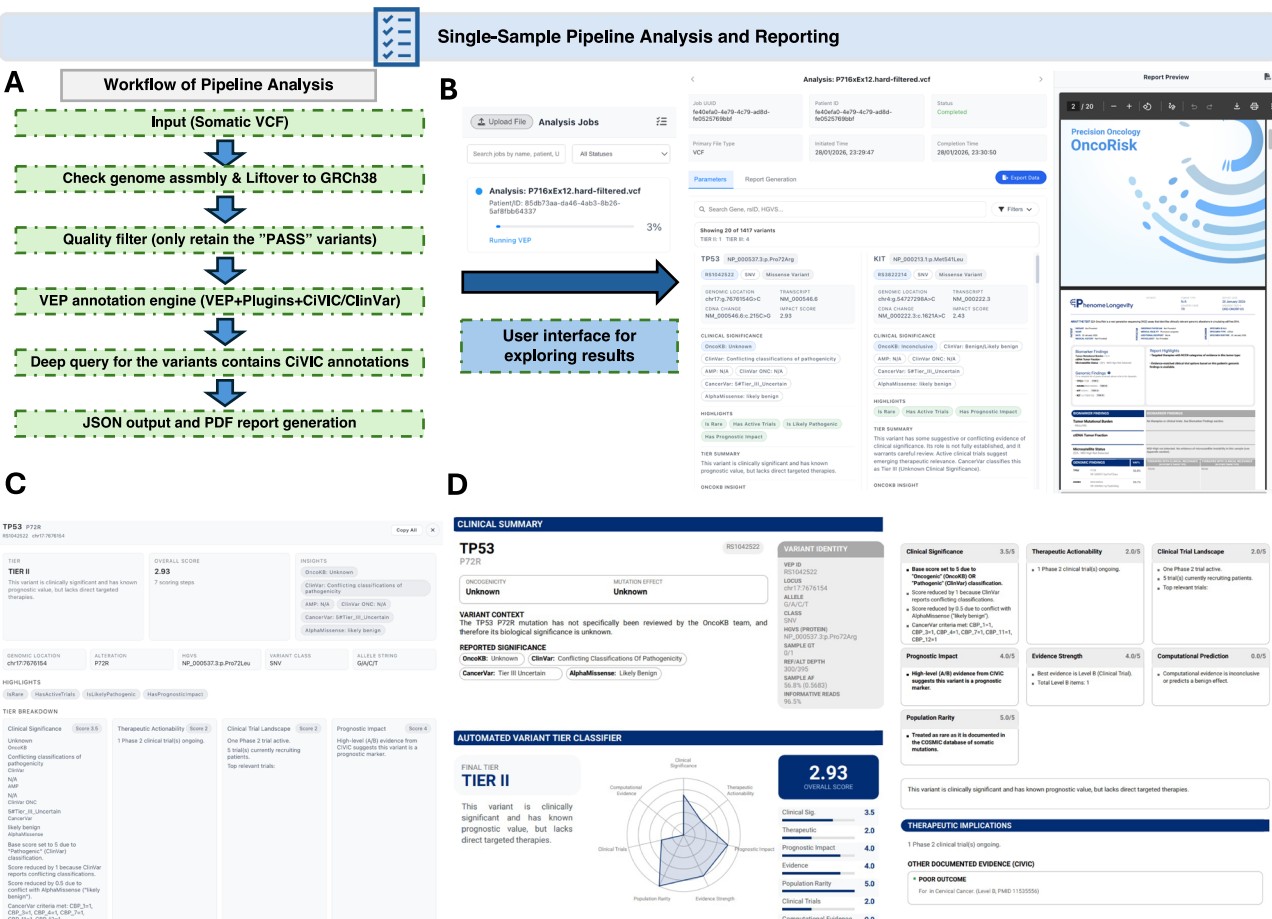

**Fig. 4 | Rapid, individualized tumor profiling and reporting via an automated workflow. A** Automated submission pipeline interface facilitating the direct upload of somatic mutation VCF files. The workflow, as indicated by the light green dashed boxes, includes an integrated liftover module for GRCh37 to GRCh38 assembly conversion and a quality control stage that filters for variants with a "PASS" flag. **B** Integrated analysis dashboard featuring a structured display of results and a live-updating PDF report preview on the right, with options for raw JSON data export. **C** OncoRisk tier assignment interface and evidence-scoring breakdown. This panel enables the presence of deep-querying results of variants against CiVIC, OncoKB, and CancerVar to assign clinical tiers 1–4 based on integrated evidence strengths. **D** Standardized clinical report output displaying graphical visualizations of the assigned tiers and therapeutic insights in the PDF format. The interface also high-lights the "filters" utility, which allows for real-time post-filtering of results based on genomic coordinates, variant consequences, and multi-database annotation tiers.

"filters" function for user directly apply post-filtering based on some key VEP annotation results in JSON files, including chromosomes, variant consequences, OncoKB annotation, CancerVar Tiers, etc.

To demonstrate the efficiency and utility of this functional module, we analyzed six tissue-derived DNA samples representing 5 different cancer types and one sample with endometriosis (Supplementary Table 1). The analysis results were shown as a demonstration on the web interface and in Supplementary Fig. 4. Through OncoRisk, we were able to annotate, query, and locate potential key findings for the uploaded samples nearly instantly. The sample with endometrial cancer (with ID P00257_DNA_1) is a rare type of cervical adenocarcinoma. The pipeline results contain 6 pathogenic (P), 5 likely pathogenic (LP), and 23 conflicting pathogenic (P/LP con-flicting with VUS), recorded in ClinVar, and 6 variants recorded in CiVIC. Focused on these 6 variants with deep queried results, we found *FGFR3* R248C (NP_000133.1:p.Arg248Cys), which was recorded both as a pathogenic and oncogenic variant, assigned as an OncoRisk tier 1 variant. This mutation is a level 1 therapeutic marker for bladder cancer treated with erdafitinib based on OncoKB, and level 4 evidence supporting the use of erdafitinib in other solid tumors. Another variant, *TP53* P72R (NP_000537.3:p.Pro72Arg), has not yet been curated in OncoKB; however, CiVIC indicates an association with cervical cancer from one of the litera-ture evidence. The sample with pancreatic cancer (P716xEx13) carried two closely mutated oncogenic (from OncoKB) mutations in *KRAS*: G12V

(NP_004976.2:p.Gly12Val), and G12R (NP_004976.2:p.Gly12Arg), which were also assigned as P/LP in ClinVar for pancreatic cancer. For therapeutic aspects, both variants are biomarkers for multiple level 1 therapies, such as avutometinib + defactinib for ovarian cancer and sotorasib for non-small cell lung cancer. The literature indicated that both mutations are well-established driver events highly associated with pancreatic cancer[22], which further confirms the results retrieved by OncoRisk. Although no targeted therapies for these variants are currently listed in OncoKB for pancreatic cancer, the CiViC data layer provides extensive supporting evidence that may inform the development of new therapeutic strategies.

**Pan-cancer-visual: pan-cancer and multi-cohort viewer for data-driven insights**

Variant-based investigations can be empowered by interrogating the genomic context of them from real-world sequencing cohorts. Based on this, we developed "pan-cancer visuals", an integrated module designed to explore the prevalence and characteristics of variants and genes across diverse pan-cancer cohorts.

We integrated somatic mutation and clinical annotation data from seven pan-cancer cohorts, including GENIE, MSK-CHORD, PCAWG, MSC, MSS, and China pan-cancer. VAFs and mutation frequencies were pre-calculated to support efficient and interactive exploration across and within cohorts.

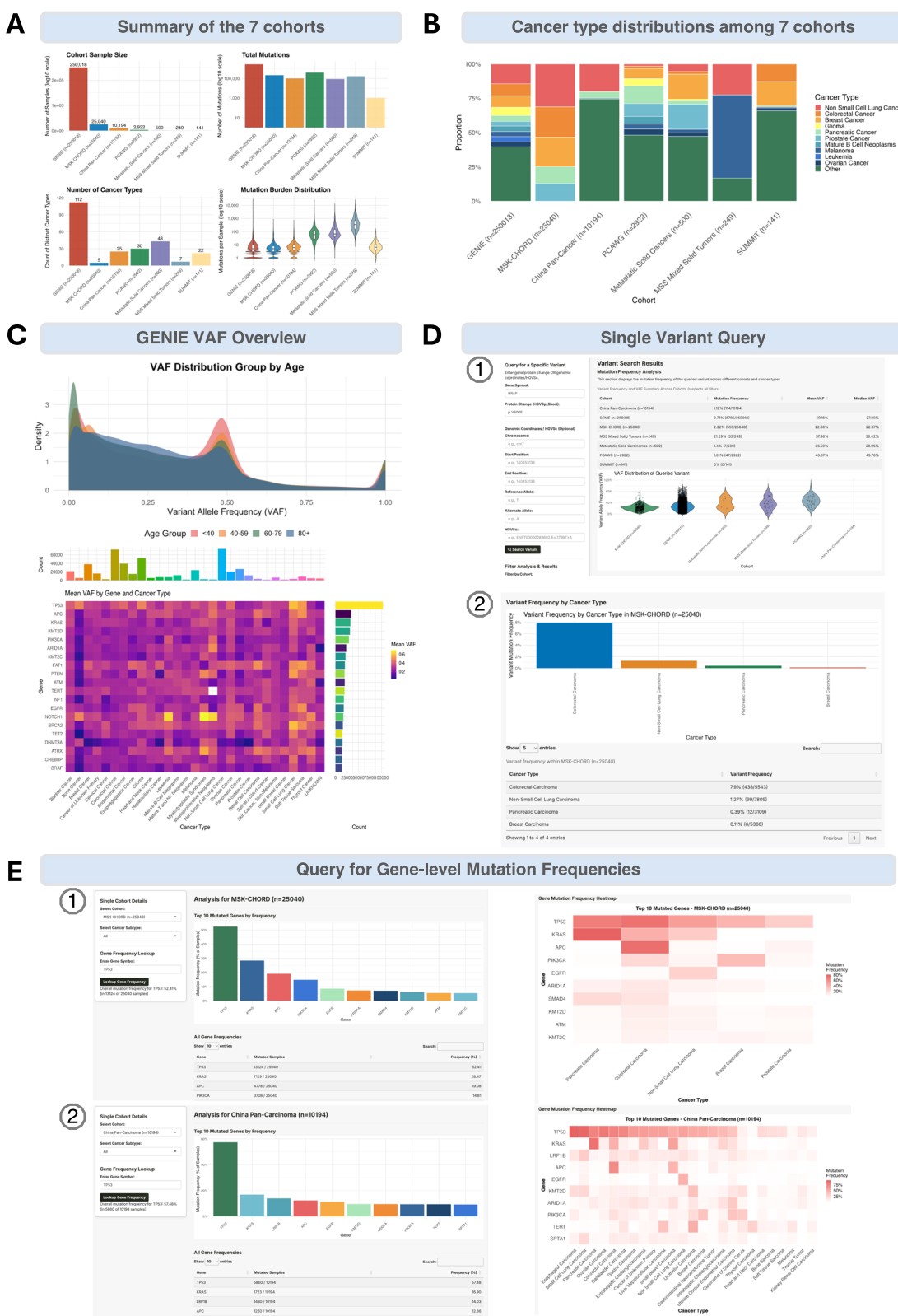

The module offers several functions. The cohort overview provides summaries of cohort size, sequencing coverage (Fig. 5A), and cancer type distribution (Fig. 5B). GENIE dominates in sample size and covers most cancer types, while MSK-CHORD provides a large sample size focused on five specific cancers. MSC, MSS, and PCAWG exhibit improved sequencing coverage, resulting in a higher number of mutations detected per sample. For cancer types being sequenced, non-small cell lung cancer currently represents the largest proportion in GENIE, MSK-CHORD, and China pan-cancer cohorts.

VAF elucidates the clonality of variants within samples and may be a crucial biomarker for individualized cancer development. Therefore, we pre-calculated the VAF for all variants across the seven cohorts. The GENIE VAF overview tab provides two complementary visualizations (Fig. 5C). A density plot shows the overall distribution of VAF across the GENIE cohort

**Fig. 5 | Pan-cancer-visual: pan-cancer and multi-cohort viewer for data-driven insights. A** Summary of cohort metrics for seven integrated datasets. The cohort sample size panel and total mutations panel display data as log10 scale. The number of cancer types panel represents the count of distinct cancer subtypes. The mutation burden distribution panel is visualized via violin plots with embedded white box-plots, where the center line represents the median. **B** Proportional cancer type distribution across cohorts. Each stacked colored segment represents one of the top 10 most prevalent cancer types identified across the entire study. All other less frequent cancer types are aggregated into a single dark green segment labeled as "other". **C** GENIE VAF overview tab representing the included GENIE dataset

($n$ = 250,018). The density plot showed the variant allele frequency (VAF) distribution grouped by age, and a heatmap summarizing the mean VAF of the top 20 most frequently mutated genes across the top 30 cancer types. The sidebar plots represent the cancer type and gene mutation distributions. The color intensity from purple to yellow indicates increasing mean VAF. **D** Single variant query tab, enabling exploration of variant-based mutation frequency and VAF distributions across cohorts and cancer types. **E** The gene analysis tab, summarizing the most frequently mutated genes in user-selected cohorts and cancer subtypes, with MSK-CHORD and China pan-cancer cohorts as examples.

---

group by age. A heatmap summarizes the mean VAF of the top 20 most frequently mutated genes across the top 30 cancer types, with sidebar plots showing the distribution of cancer types and mutation frequencies per gene. We also pre-calculated all variant-based and gene-based mutated frequencies, accordingly, describing the prevalence of a given variant or gene in a selected population. The single variant query tab allows users to examine variant-based mutation frequency, mean/median VAF values, and VAF distribution within selected cohorts and cancer types (Fig. 5D). These characterization of gene-level mutation frequencies across cancer subtypes provides a foundation for the discovery of novel biomarkers.

The gene analysis tab summarizes the most frequently mutated genes within user-selected cohorts and cancer subtypes, illustrated here with MSK-CHORD and China pan-cancer cohorts as examples (Fig. 5E). From this view, we identified well-known pan-cancer mutations in *TP53* and *KRAS* across most cohorts, and *ERBB2* specifically within the SUMMIT cohort.

### Server-maftools: comprehensive somatic mutation analysis tool set for cancer cohorts

Server-Maftools can be initiated by either uploading user-provided mutation files (TXT, MAF, or TSV) or by selecting from preloaded datasets (Fig. 6A). A total of 33 TCGA cohorts and 2427 CCLE cell line profiles (DepMap 2024 Q2)[23] were incorporated as ready-to-use resources for the load and analysis page. For CCLE, text-based filtering supports subsetting by clinical metadata, such as tumor type or data type. Uploaded datasets are automatically summarized and prepared for downstream analysis. All functionalities are listed in Fig. 6B, including three sub-tabs (cohort visualizations, gene/sample visualizations, and cohort analysis) as explained in detail below.

### Cohort-based visualizations for the general mutation landscape.

Mutation landscapes can be explored through summary plots, oncoplots, Ti/Tv analysis, mutation load comparisons, and recurrent mutation detection. The mutation summary plots, oncoplots, and Ti/Tv plots can be generated after loading or uploading MAF files. From the mutation load comparison section, the analyzed cohort can be directly compared with the full TCGA cohort. Customized functions were made for oncoplots that allow grouping by pathways (sigpw, known oncogenic signaling pathways; or smgbp, pan-cancer significantly mutated genes classified by biological process). As an illustrative case, the TCGA-KIRC cohort (kidney renal clear cell carcinoma; $n$ = 370) was analyzed. The oncoplot showed that *VHL*, *PBRM1*, *TTN*, *SETD2*, and *BAP1* were the top five mutated genes, which is consistent with current knowledge of KIRC-specific mutation characteristics (Fig. 6C). A recurrent mutation plot was added to visualize recurrently mutated variants (based on a user-defined threshold), showing that the *VHL* gene contains 64 unique variants that occurred at least twice (Fig. 6D).

### Genes/samples-based visualization.

Interactive gene-level views include lollipop plots, rainfall plots, and VAF distributions. Beyond the standard maftools outputs, we implemented theme-adjustable lollipop plots using G3Viz, exemplified by *VHL* mutations in the KIRC cohort (Fig. 6E). Clicking individual variants retrieves detailed annotations. The rainfall plot function identifies samples with kataegis. While no such

events were detected in the KIRC cohort, analysis of the LAML cohort successfully identified multiple samples exhibiting these hypermutation patterns. VAF distribution plots are plotted alongside for help with estimating the clonal status of the top mutated genes.

### Statistical analysis for initial understanding of the somatic mutation profiles.

A total of eleven sections were developed for the cohort analysis tab, which integrates eleven modules adapted from maftools with adjustable parameters. Somatic interaction networks reveal co-occurrence and mutual exclusivity patterns between gene mutations within the cohort, and results were visualized through plots and tables summarizing statistical analysis. We analyzed the somatic interaction pattern of TCGA-KIRC ($n$ = 370), showing that VHL significantly co-occurred with *PBRM1* ($P$ = 0.011, OR = 1.74), while PBRM1 was exclusively mutated with *BAP1* ($P$ = 0.005, OR = 0.31), and co-occurred with *SETD2* (Fig. 6F). Mutation-based survival analysis with interactive gene selection enables the fast exploration of the prognostic implications of specific mutations within a group of gene sets. *VHL* mutant/wild type showed no direct significance in patient prognosis ($p$ = 0.739, HR = 1.07), but *BAP1* alone showed significant prognostic effects ($p$ = 0.0148, HR = 1.87). Grouping *BAP1*, *KDM5C*, and *MTOR* ($n$ = 84) further strengthened the statistical significance ($p$ = 3e-4, HR = 2.07) compared to the wild-type group ($n$ = 286) (Fig. 6G). Besides these, we also employed this module for cancer driver gene identification, applied the Onco-driveCLUST algorithm, and detected *TCEB1* as a significant driver gene in KIRC. Protein Pfam domain impact analysis in KIRC revealed the 7tm_1 domain (GPCR family) as the most frequently mutated domain with 327 mutations across 237 genes, followed by the Zn-finger domain (COG5048) with 229 mutations across 166 genes, both reflecting widespread background mutational events.

Two functional modules, mutational comparison and clinical enrichment, were developed to facilitate the identification of differentially mutated genes and clinically enriched alterations across user-loaded cancer cohorts. In the mutational comparison module, analysis of KIRC and KIRP ($n$ = 282) cohorts revealed distinct mutation profiles. Specifically, *VHL* ($P$ < 0.001, KIRC $n$ = 172, KIRP $n$ = 3, OR = 80.51, 95% CI: 26.44–398.82), *PBRM1* ($P$ < 0.001, KIRC $n$ = 148, KIRP $n$ = 12, OR = 14.95, 95% CI: 8.03–30.37), and *MTOR* ($P$ < 0.001, KIRC $n$ = 31, KIRP $n$ = 4, OR = 6.34, 95% CI: 2.20–25.03) were significantly enriched in TCGA-KIRC. Conversely, *MET* ($P$ < 0.001, KIRC $n$ = 3, KIRP $n$ = 22, OR = 0.10, 95% CI: 0.02–0.33) and *OBSCN* ($P$ < 0.001, KIRC $n$ = 5, KIRP $n$ = 21, OR = 0.17, 95% CI: 0.05–0.47) showed significant enrichment in TCGA-KIRP (Fig. 6H). In the clinical enrichment module, stratification of mutations by CDR-defined AJCC pathological tumor stage under a manually set significance threshold of $p$ < 0.02 demonstrated that *SETD2* (13/48, $P$ = 0.005, OR = 2.94, 95% CI = 1.30–6.36) and *BAP1* (10/48, $P$ = 0.018, OR = 2.75, 95% CI = 1.10–6.42) were significantly more frequently mutated in stage IV patients compared with the remaining stages (Fig. 6I).

Other functions not shown as examples include tumor mutational burden assessment, which calculates the median TMB value and the TMB distribution across the cohort. Drug–gene interaction mapping links mutation profiles to pharmacogenomic targets. Oncogenic pathway analysis reveals pathway-level disruptions with enrichment statistics and visual maps. MATH analysis estimates intratumor heterogeneity from VAFs.

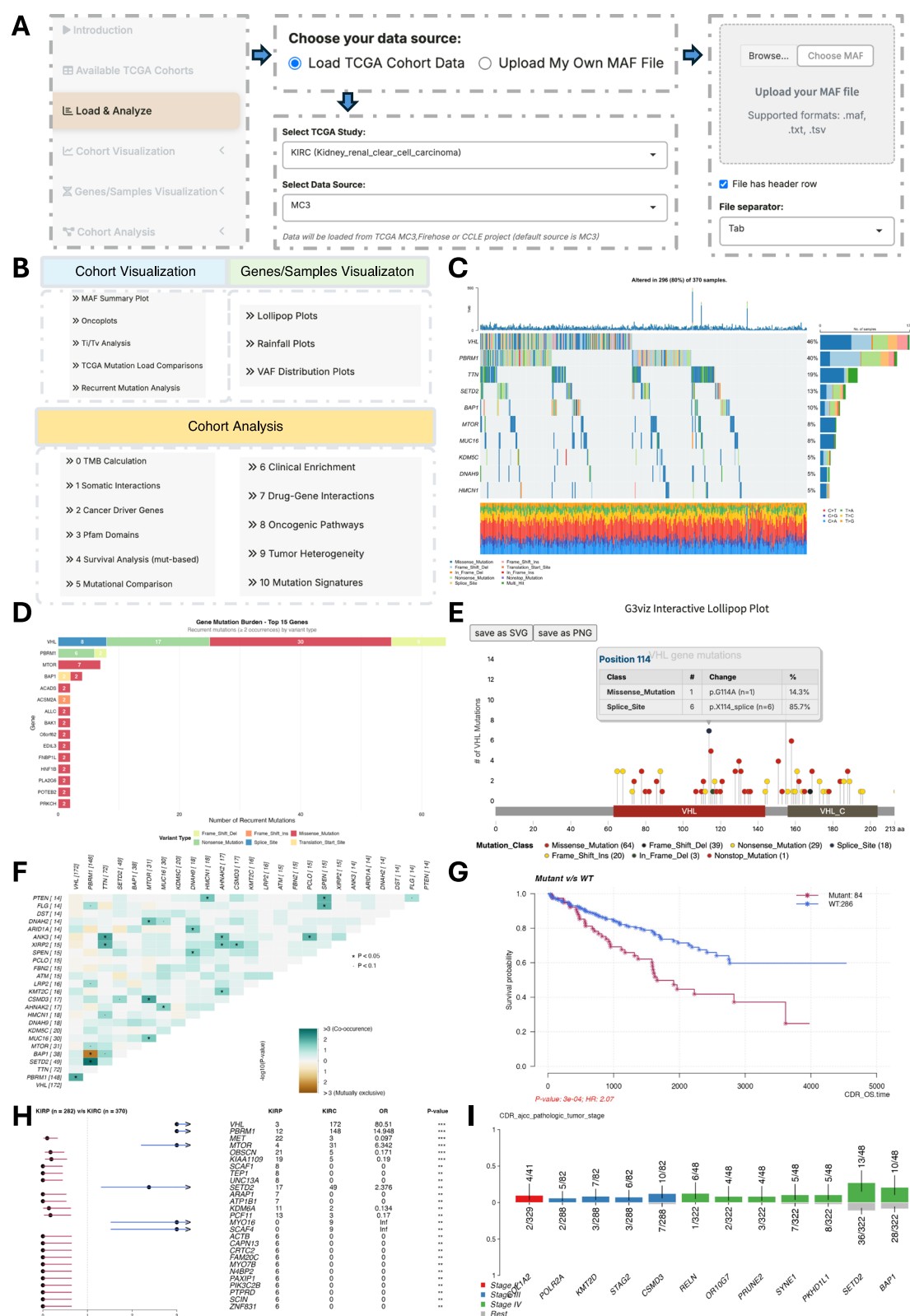

Mutational signature analysis extracts mutational processes and compares them with COSMIC reference signatures.

## Discussion

The importance of integrative precision oncology platforms has been widely recognized. Suehnholz et al.[24] emphasized the importance of building POKBs that incorporate therapeutic (Tx), diagnostic (Dx), and prognostic (Px) insights. Stenzinger et al.[25] then highlighted the major challenges in the routine clinical utility of tumor sequencing, including long diagnostic turnaround times and the limited capacity of the healthcare workforce to integrate molecular profiling results into clinical practice. These discussions emphasize the urgent demand for web-based POKBs.

OncoRisk especially focuses on linking knowledge-based databases that facilitate clinical oncology decisions following AMP/ASCO[26] and

**Fig. 6 | The interactive analysis interface and functional overview of server-maftools. A** Workflow for initiating an analysis, showing options to load pre-processed TCGA/CCLE datasets or upload user-provided mutation files. **B** The navigation panel lists all available visualization and analysis modules categorized into three main tabs: cohort visualization (the light blue box), genes/samples visualization (the light green box), and cohort analysis (the light orange box). **C** An oncoplot generated from the TCGA-KIRC cohort (*n* = 370), highlighting the top mutated genes, including *VHL* and *PBRM1*. **D** Recurrent mutation plot for the KIRC cohort, showing the distribution of recurrent variants in the *VHL* gene. **E** Interactive lollipop plot displaying *VHL* mutations, styled with a cBioPortal theme. **F** Somatic interaction network analysis for the TCGA-KIRC cohort used pairwise Fisher's exact tests to reveal significant co-occurrence (e.g., *VHL* and *PBRM1*, *P* = 0.011, OR = 1.74) and mutual exclusivity (e.g., *PBRM1* and *BAP1*, *P* = 0.005, OR = 0.31) patterns. **G** Kaplan–Meier survival curves demonstrating the prognostic impact of single-gene (*BAP1*) and multi-gene (*BAP1, KDM5C, MTOR*) mutation signatures. Patients with mutations in *BAP1, KDM5C*, or *MTOR* (*n* = 84) showed significantly shorter overall survival compared to the wild-type group (*n* = 286; log-rank *P* = 3e-4). The calculated hazard ratio (HR) was 2.07 (95% CI: 1.38–3.11). **H** Results from the mutational comparison module, identifying genes differentially mutated between TCGA-KIRC and TCGA-KIRP (*n* = 282) cohorts using Fisher's exact test. The horizontal error bars represent 95% confidence intervals (CI). The plot used *P* = 0.01 as the cutoff. Genes including *VHL* (*P* < 0.001, KIRC *n* = 172, KIRP *n* = 3, OR = 80.510, 95% CI: 26.440–398.816), *PBRM1* (*P* < 0.001, KIRC *n* = 148, KIRP *n* = 12, OR = 14.948, 95% CI: 8.033–30.373), and *MTOR* (*P* < 0.001, KIRC *n* = 31, KIRP *n* = 4, OR = 6.342, 95% CI: 2.204–25.026) were significantly enriched in the KIRC cohort. *MET* (*P* < 0.001, KIRC *n* = 3, KIRP *n* = 22, OR = 0.097, 95% CI: 0.018–0.328) and *OBSCN* (*P* < 0.001, KIRC *n* = 5, KIRP *n* = 21, OR = 0.171, 95% CI: 0.050–0.473) showed significant enrichment in the KIRP cohort. **I** The bar plot illustrates the enrichment of specific gene mutations across different AJCC pathologic tumor stages in the TCGA-KIRC cohort, using the *P* = 0.02 as the cutoff for plotting. For each gene, the colored bars (red for stage I, blue for stage III, green for stage IV) represent the mutation frequency at that stage, while the gray bars represent the combined mutation frequency across all other stages (rest). Statistical significance was determined using a group-wise two-sided Fisher's exact test. For example, SETD2 (13/48, *p* = 0.005, OR = 2.94, 95% CI = 1.30–6.36) and BAP1 (10/48, *p* = 0.018, OR = 2.75, 95% CI = 1.10–6.42) mutations show significant enrichment in stage IV tumors compared to the rest of the cohort.

ESMO guidelines[14] via a series of quick and strong query functions. We showed that OncoRisk integrates these resources to map mutations within multi-layer information, including structural, functional, and clinical contexts, only by simple and fast query functions. The network visualization module goes beyond conventional tab-based designs by integrating multi-term search results into a single interactive network, thereby promoting a more comprehensive and dynamic exploration for user-defined queries, as demonstrated by our *EGFR* case study.

The constructed unified cancer family trees enhance the representation of cancer relationships and have combined the advantages of both DO and OncoTree classification systems to link more nodes (specific cancer sub-types), offering additional opportunities to identify targets and therapies that may be shared among related (ancestor/descendant) cancer types. Furthermore, the visualization of drugs within protein pockets alongside associated variants provides direct visualization to infer potential mechanisms of drug action or resistance based on spatial proximity.

The individualized tumor profiling and reporting pipeline is another important innovation of OncoRisk, which fills in the current limitation of such applications. These individualized genomic markers are one of the key components for personalized modeling using multi-omics layers for precision medicine[27]. It enables the rapid generation of patient-specific reports by parallel analysis of thousands of somatic mutations, integrating knowledge bases to identify potential drug targets and relevant clinical trials. This system can serve as a copilot to comprehensive genomic profiling (CGP) assays, such as elio tissue complete CGP[28] and Illumina TSO 500 CGP, supporting the full process from tissue to report, which helps clinicians act on molecular profiling results[29]. This is also an advance compared to another oncogenic reporting platform, PORI[30], that included patient reporting modules but required complex local installation and configuration with limited database availability. We believe that this designed PDF-based format, following ESMO recommendations[26], will be beneficial for patients to understand their own tumor cases and for clinicians to make informed decisions.

OncoRisk's utility is further amplified by its flexible pan-cancer data ecosystem. We proposed the idea that using existing large-scale, real-world pan-cancer cohorts not merely as static references but as dynamic standards for hypothesis generation and validation, which will facilitate rapid access to general mutation features, thereby helping studies, such as the pan-cancer characterization of individual genes[31]. While established platforms like cBioPortal have democratized data access, OncoRisk enables deeper, more complex analyses—such as pan-cancer clinical parameter and gene frequency plots—within a streamlined workflow. It also provides a more comprehensive and customizable toolset than platforms like Onkopus[32], Tri©DB[33], PANDA[34], and Oviz-Bio[35], which lack integrated workflows for complete patient profiling and extend beyond the focused scope of research-oriented servers like OncoVar[36]. Eventually, OncoRisk can serve as a valuable tool for population-scale initiatives that generate massive genomic data[37], leveraging its ability to rapidly summarize and analyze large cohorts and translating these insights into personalized oncology. In summary, we systematically compared OncoRisk with over 10 other databases/servers to emphasize the features of OncoRisk (Supplementary Table 3).

Despite its many strengths, OncoRisk has several limitations that we aim to address in future work. Complex genomic events, such as amplifications and gene fusions, are not yet included in the reporting pipeline, since they are typically generated from sources other than the small-variant VCF in tumor sequencing. However, users can still access relevant information through manual queries. While OncoRisk's pipeline analysis provides a quick approach for initial evidence synthesis on clinical usage, it is currently positioned as a web tool helping the process rather than a fully practical diagnostic system. The Tier and scores calculated by OncoRisk represent the information that can be gathered; it is not intended to replace the standard AMP/ASCO interpretation guidelines. The VEP annotation runs with a fixed setup, which future work may aim to add more flexibility to this process. In addition, the current 3D visualization of proteins, variants, and drugs with supporting evidence does not yet provide predictive analyses of how mutations alter protein–drug interactions from the reference allele to the alternative allele. Future work will aim to integrate algorithms from webservers specifically designed for mutation–protein–drug interaction analysis, such as Swiss-PO[38] and MutationExplorer[39]. Another future direction of OncoRisk is to integrate data on the other biological layers that impact cancer development, especially the transcriptome and methylome[40].

## Methods

### Database integration and exportation

OncoRisk has integrated multiple resources to achieve comprehensive functions in all modules, as summarized in the Supplementary Table 2 and Supplementary Fig. 5.

The data exploration layer integrates CiViC[10,41], OncoKB[12], ClinVar[9] (2025 version), CancerVar[13], and ClinicalTrials.gov through a MySQL-based backend system. Specifically, we utilized CiViC as the foundational ontology for defining oncogenic categories and constructing the knowledge base. We selected this resource for its open-source, highly structured nature and its unique capacity to map diverse clinical biomarker nomenclatures (e.g., variant groups and fusions) to specific genomic alterations. Then, we complemented it with specific pathogenicity and drugability data from other databases. These data are accessed via programmatic APIs and modularized into relational entities (genes, variants, therapies, diseases, and evidence) with indexed schemas for efficient queries and reuse. To enhance response times and system efficiency, frequently accessed data entries were cached in Redis and precomputed term-association networks. External API

calls are managed using batch request pooling and exponential backoff retry mechanisms to handle rate limits.

Input variant queries, including single-variant queries or uploaded patient sequencing files, are standardized through HGVS and rsID mappings and executed via parameterized SQL for safe and consistent retrieval. Entity relationships are precomputed as adjacency lists and passed forward for visualization and tiering pipelines.

All processed data (oncogenic terms in knowledge bases, deep query results, and pipeline analysis outputs) is made exportable in JSON formats. This also enables integration with PDF clinical-reporting modules, front-end exploration views, and external research pipelines. OncoRisk also provides a structured RESTful API that enables programmatic access to its integrated knowledge base across Windows, Linux, and macOS environments.

## Data resources in pan-cancer-visual and server-maftools

A total of seven curated sets of nonredundant pan-cancer cohorts were downloaded and used in OncoRisk for calculating gene-based, variant-based mutation frequencies and VAFs. MSK-CHORD ($n = 25040$), Metastatic solid cancers ($n = 500$), MSS mixed solid tumors ($n = 249$), SUMMIT-neratinib basket study ($n = 141$), China pan-cancer ($n = 10194$), PCGWA ($n = 2922$) were downloaded directly from cBioPortal[20], while the GENIE dataset ($n = 250,018$, public release version 18.0) was downloaded from GENIE synapse[19]. Each cohort included MAF format somatic mutation files and clinical data.

TCGA cohorts and CCLE cell line data (2024Q2) were retrieved using the R packages maftools[42] and TCGAmutations, with harmonized clinical data resources[43]. Server-maftools maintains a reactive object that serves as the active dataset for all downstream analyses, with uploaded data automatically replacing pre-loaded cohorts and dynamically updating all analysis parameters (gene lists, sample selections).

## In-house individual tumor tissue sequencing

Tumor FFPE tissues are sliced with a microtome after marking the regions with tumor cells. DNA is isolated manually from the FFPE slices with QIAamp FFPE DNA kit (QIAgen, Hilden, Germany) or automatically using Maxwell RSC FFPE plus DNA kit (Promega, Madison, WI, USA). For the extractions, the protocols provided by the manufacturers are followed. After the measurement of the amount of DNA fluorometrically with Qubit (Waltham, MA, USA), the quality check of DNA samples is performed by quantitative PCR with Illumina FFPE QC kit (Illumina, San Diego, CA, USA). Samples with $\Delta Cq \leq 5$ are considered suitable for library construction.

The NGS libraries are constructed with TruSight Oncology 500 v2 panel kit (Illumina, San Diego, CA, USA). The library quality and quantity are checked prior to PE $100 \times 2$ sequencing on the NovaSeq 6000 platform. Each library generates at least 10 GB of data. After demultiplexing and BCL-to-FASTQ conversion using Dragen v4.3 (Illumina, San Diego, CA, USA), the pipeline performs alignment and variant calling to generate VCF files. These variants are subsequently filtered and annotated using Illumina Connected Insights and then uploaded to OncoRisk for analysis.

## OncoTree and disease ontology (DO) for constructing the disease family tree and grouping cancers by tissue

Disease query terms were mapped from the disease ontology (DO)[44] database to OncoTree[45] tissue layers through hierarchical traversal. First, each query was mapped to specific DO graph edges, with synonym-based normalization applied. Next, string matching was performed between the normalized DO terms and the OncoTree nodes. If no match was found, the process proceeded by traversing to the parent concept of the current DO term, after which a new string-matching attempt was made. This iterative progression from specific (child) to general (parent) concepts within the DO hierarchy continued until a match was identified in OncoTree. Once a match was identified, tissue classification was determined by traversing OncoTree parent nodes upward to tissue-level categories. Lastly, the

network was constructed with combined family tree information to link all query diseases to their DO subtypes, and further to the tissue-based cancer families from OncoTree.

## Download of RCSB PDB structures and visualizations

Protein structure files in PDB format were downloaded from the PDBe repository (RCSB PDB)[46] via rsync and saved locally as .ent files. Related identifiers of the crystal structures of protein-drug combinations were retrieved using the REST API with search functions and parsed to extract protein chain coordinates, small molecule ligands, and structural metadata. Mutation sites were mapped to structural coordinates using standard amino acid substitution notation (e.g., N549H). Three-dimensional visualization was implemented using 3Dmol.js[47] to generate interactive HTML outputs. Protein chains were rendered as cartoon models, while small molecules were displayed as stick representations. Mutation sites were highlighted using combined sphere and stick models with positional labels, with flanking residues (±2 positions) shown for structural context.

## LiftOver

The input VCF files were first used to determine the genome assembly by examining VCF headers or by checking chromosome lengths. VCF files aligned to GRCh37/hg19 were liftovered to GRCh38 coordinates using CrossMap (v0.6.5)[48] with the UCSC hg19ToHg38.over.chain.gz chain file and GRCh38 reference genome. Only PASS-filtered variants were retained for subsequent analysis.

## VEP annotation

Variant annotation was performed using the Ensembl Variant Effect Predictor[49] (VEP, v113) with GRCh38 assembly. The workflow receives users' input from front-end input as single variant identifiers or from VCF/query files uploading, automatically detects input format, and applies necessary PASS filtering and liftover for complete VCF input. We utilize the --pick flag with a defined order: "--pick_order mane_select, rank" to prioritize MANE select transcript when multiple RefSeq isoforms are available for a single variant. The annotation also incorporates HGVS/HGVSG nomenclature, variant classification, canonical transcript selection, population allele frequencies from gnomAD exome and genome datasets, clinical annotation database ClinVar (20250623), and CIViC (20250630). Functional impact predictions were integrated from in-silico tools, including REVEL, dbscSNV, BayesDel, SpliceAI, and AlphaMissense. Results were output as compressed JSON files for further processing and manual adjustment options, interactive with the front-end.

## The clinical reporting pipeline and the tier attributing system

Within the clinical reporting workflow, uploaded sample VCF files undergo variant effect prediction (VEP) annotation with custom annotation sources for all variants presented. Then, to ensure server stability and comply with external API rate limits, an initial filtering step is applied by default to prioritize variants containing evidence strings from CIViC. This pre-selected variant set subsequently undergoes the same deep annotation and analysis pipeline as individual variants submitted via the deep query function. An adjustable filter module allows the user to re-adjust the variants that go into deep-query and tier attributing.

A rule-based variant classification system was developed that assigns each variant a final clinical tier (I–IV) based on quantitatively integrated evidence from multiple sources (Supplementary Data 1). The system employs clinical significance, therapeutic actionability, prognostic impact, evidence strength, computational prediction, and population rarity as logical rules to calculate the summary tier scores of the variant. The clinical significance score integrates oncogenic assertions from OncoKB and ClinVar with adjustments from AlphaMissense predictions. The therapeutic actionability score evaluates drug sensitivity and resistance evidence from OncoKB and CiVIC, prioritizing FDA-approved therapies. The prognostic impact and evidence strength scores are derived from CiVIC evidence levels. The computational prediction score incorporates AlphaMissense, REVEL,

SIFT, and PolyPhen-2 predictions. The population rarity score assesses variant frequency using gnomAD for germline variants and OncoKB/COSMIC for somatic variants.

## Calculations and visualizations in the pan-cancer-visual

The computations required for pan-cancer-visual are pre-cached in the local dataset to ensure efficient query performance. Specifically, we derived the following measures: VAF, defined as the fraction of sequencing reads supporting a variant within a single sample, calculated as VAF = alternative allele count/total allele count, and filtered for a valid interval [0, 1]. Variant mutation frequency represents the prevalence of a specific variant (e.g., *BRAF* V600E) across a sample population. For a selected cohort and cancer type, it is calculated as the number of unique samples with the variant divided by the total number of unique samples in the group. Similarly, gene mutation frequency was calculated as the number of unique samples with any mutation in the gene divided by the total number of unique samples in the group.

## Statistics and reproducibility

Statistical analyses for tumor cohorts were performed using R and the maftools package, as represented by the analysis for the TCGA-KIRC ($n = 370$) and TCGA-KIRP ($n = 282$) cohorts. Two-sided Fisher's exact tests were used for mutational comparisons, clinical enrichment, and somatic interaction analyses. Associations were quantified using odds ratio (OR) and 95% confidence intervals (CI) reported to three decimal places. Survival curves were generated using the Kaplan–Meier analysis and compared via log-rank tests. Hazard ratios (HR) and 95% CI were calculated using Cox proportional hazards models. A *P* value of less than 0.05 was considered statistically significant unless otherwise stated. Reproducibility was ensured by using standardized bioinformatics pipelines on publicly available datasets. No experimental replicates were performed as this study is based on retrospective clinical cohorts.

## System architecture and user interface

The web architecture combines modular data representation, indexed storage, caching, and synchronized API integration to support fast and reusable variant-centered exploration.

The main user interface of OncoRisk was built with React and related JavaScript frameworks, using DOM manipulation and component-driven design. Charts, 3D models, and organ visualizations were natively developed, while network visualizations leveraged NPM packages. The backend runs on Node.js with middleware managing cookie-based authentication, rate limiting, request tracking, and secure data uploads that are authenticated, encrypted, and chunked. Requests are throttled and optimized through an in-memory caching layer, with frequently traversed GraphQL paths cached to avoid recomputation, while asynchronous endpoints query databases only when fresh data is required. Database queries are aggressively optimized with indexing strategies, and all data is normalized into SQL for seamless integration, interoperability, and exporting.

Specialized analysis modules (server-maftools and pan-cancer-visual) were built in R Shiny, with reactive programming controls data flow, updating selectable genes, samples, and clinical features after input. Mutation summaries were generated with maftools and rendered as interactive tables (DT). Visualization modules employed maftools, plotly, ggplot2, g3viz, and pheatmap for mutation plots, heatmaps, survival, and pathway analyses. Signature extraction used NMF. Deployment was configured with rsconnect::deployApp for cross-platform and browser compatibility.

## Computational performance and scalability

Computational pipelines are batched and queued to optimize resource usage, with parallel threading accelerating analysis workloads. Pipeline VCF analysis is executed via child processes, and job execution is monitored through encrypted queuing tables for full auditability. Natively developed PDF generation systems follow the same queuing model. Users are recommended to upload somatic mutation profiles only, with a maximum size of 75 MB. To ensure high concurrency, the system utilizes 16 parallel workers and SQL connection pooling, achieving a throughput of approximately 15,000 somatic VCF files (each around 4–6 MB per file) per day. To ensure low-latency interactive performance, frequently accessed database partitions are indexed and resident in RAM for hot in-memory data serving, minimizing disk I/O. Furthermore, optimized SQL connection pooling enables high levels of concurrent access without contention, maintaining predictable performance even under increasing user load.

Server-maftools and pan-cancer-visual are hosted on the server that allocates 2vCPU and 4 GB of memory/RAM. For Server-maftools, uploaded MAF files (≤100 MB) were processed into MAF objects through maftools with validation routines for required data columns.

The infrastructure integrates Cloudflare-backed network proxies for security, and CI/CD pipelines with containerized deployments ensure reliability and zero-downtime updates. Orchestration is handled through container management and scaling systems, with observability layers for monitoring, logging, and automated recovery. Structured APIs are provided, which unify communication across components.

## Data robustness and privacy

For the uploaded files or search inputs that do not fully conform to the expected formats, the system still accepts and processes them to maintain robustness. However, these incorrectly formatted or invalid inputs may lead to incomplete analyses or the absence of returned results, depending on the nature of the deviation.

To ensure maximum data privacy and security in pipeline analysis, all user-uploaded files are processed according to strict protocols. Each upload is anonymized using a universally unique identifier (UUID), and data is encrypted both in transit and at rest using industry-standard security protocols. Uploaded datasets are strictly used for the active analysis session and are never sold, shared, or accessible to any third parties. To prevent unauthorized data retention, all files are automatically purged from our secure server after 30 days.

## Reporting summary

Further information on research design is available in the Nature Portfolio Reporting Summary linked to this article.

## Data availability

The OncoRisk web server is freely accessible at https://www.phenomeportal.org/oncorisk. The six tumor sequencing files (VCF format, tumor DNA sequencing) and the full analyzed outputs (CSV format) used for the validation of the OncoRisk pipeline analysis module have been provided in Zenodo with https://doi.org/10.5281/zenodo.19069148[50]. The Source data for Figs. 5A–C and 6D were provided in Supplementary Data 3. The pan-cancer cohort datasets analyzed in this study are publicly available via cBioPortal, with the exception of the AACR Project GENIE dataset (v18.0), which was accessed via Synapse under application ID ACT-2119. TCGA and CCLE data were retrieved using the TCGAmutations and maftools R packages. Clinical knowledge data (including CiVIC, ClinVar, and CancerVar) were accessed via their public APIs or bulk VCF downloads, while OncoKB data were accessed via API under an approved academic license. All other data supporting the findings of this study are available within the article and its Supplementary Information.

## Code availability

The source code for the OncoRisk web server is hosted on GitHub and is publicly accessible at: https://github.com/xiyasong/OncoRisk_Codes with the release v1.0.0 under the Zenodo https://doi.org/10.5281/zenodo.18984617[51].

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

## Acknowledgements

This work is funded by the Knut and Alice Wallenberg Foundation, Sweden (project number 72110 to A.M.), and Phenome Longevity, Inc., USA. We also acknowledge the American Association for Cancer Research (AACR) for the AACR Project GENIE registry, and thank cBioPortal, CIViC, OncoKB, ClinVar, and CancerVar for providing public data access. We also acknowledge the developers of the TCGAmutations and maftools R packages.

## Author contributions

X.S.: conceptualization, methodology, software, formal analysis, investigation, data curation, visualization, writing-original draft. E.G.: methodology, software, data curation, visualization, writing-review and editing. X.L.: data curation, writing-review and editing. H.T.: data curation, writing-review and editing. G.Y.: data curation, writing-review and editing. B.Y.: data curation, writing-review and editing. M.U.: writing-review and editing. C.Z.: writing-review and editing. A.M.: conceptualization, supervision, methodology, project administration, funding acquisition, writing-review and editing. All authors read and approved the final manuscript.

## Funding

## Competing interests

The authors declare the following competing interests: A.M. is a co-founder of Phenome Longevity Inc. G.Y., B.Y. are employees of Phenome Omics R&D. The remaining authors declare no competing interests.

## Additional information

**Supplementary information** The online version contains Supplementary material available at https://doi.org/10.1038/s42003-026-10005-5.

**Peer review information** : *Communications Biology* thanks Daoud Meerzaman, Shimoga Chandrashekar Darshan and the other anonymous reviewer(s) for their contribution to the peer review of this work. Primary handling editors: Dr. Nilanjan Banerjee. A peer review file is available.

