## [Transparent Peer Review File · Communications Biology]

OncoRisk: A state-of-the-art Web Server for bridging the oncogenic databases and pan-cancer cohorts to the translational oncology

Corresponding Author: Professor Adil Mardinoglu

Version 0:

Reviewer comments:

Reviewer #1

(Remarks to the Author)

Review of "OncoRisk: A state-of-the-art Web Server for bridging the oncogenic databases and pan-cancer cohorts to the translational oncology" by Song, et al.

The authors present OncoRisk, a server that integrates data from 10 oncogenic databases and seven pan-cancer cohorts, facilitating easy exploration of genomic variants, gene interactions, and therapy associations. It provides automated generation of a variety of tables, plots, and reports, enabling rapid assessment of various mutations. The server is very useful, enabling rapid exploration of a variety of cancer-related associations ranging from individual mutations to pan-cancer cohorts. While most, if not all, of the functions provided can be performed by other servers or software tools, it appears that OncoRisk uniquely integrates the specific combination of data sources and analysis functions available in a very user-friendly interface. It is likely to be a highly useful tool for the community.

My concerns have to do with the lack of transparency about the site it is hosted on. Phenomeportal.org appears to be owned by the Align Institute, a non-profit organization dedicated to building large, open biological datasets. However, this is not clearly explained on the phenomeportal.org website, nor is phenomeportal.org explicitly referenced on the Align website (alignbio.org). Phenomeportal.org hosts the XOMics portal, which appears to be related to szaportal.com (which redirects to phenomeportal.org). The Documentation button on the main page directs to <https://docs.szapfs.org/oncorisk/>, which does not load. (I was unable to get the szaomics.com website to load.) Two of the manuscript authors are affiliated with SZA Omics R&D. Other tools on phenomeportal.org (e.g. GenRisk) appear to be publicly available, but the XOMics Portal requires an email to log in. Furthermore, there is no explanation that I could find regarding the privacy protocols for uploaded files. Clearly a major beneficial feature of OncoRisk would be the ability to upload one's own VCF or MAF files, but without clarity on who owns the website and what, if any, privacy rules apply to the uploaded data, most users will not be able or willing to use this functionality. Clearly the server is useful as-is for exploring public oncogenic databases. However, the lack of clarity about the relationships between the server website, the Align Institute, the SZA Omics corporation, and the manuscript authors raise questions about the ability to upload user data and the long-term availability of the server.

I would like to see more clarity about the relationships between the authors, the Align Institute, and the SZA Omics corporation, as well as any pertinent commercial products related to OncoRisk, but I would not say such explanations are required for publication as the site is currently available and functional. More clarity about what happens to data that is uploaded, even if it is warnings that privacy cannot be guaranteed, would be highly desirable.

Minor Issues:

The buttons on the main server website are confusing. They react when you mouse over them, but nothing happens when you click on them. There seems to be different categories of buttons, however. Clearly the buttons in the Quick Actions panel are meant to do something when clicked, but none of them work. It's not clear that anything is meant to happen when you click on the buttons in the Knowledge Base Overview or Latest Updates panels. If those are meant to just be static descriptors they should not change when you mouse over them.

The Help & Support button on the main page states that you can "Access documentation, tutorials, and contact technical

support”, but the button just takes you to the 7-minute video overview which is hosted on Google Drive (i.e. it’s not on a publicly available YouTube channel). The Settings button states that you can “Configure application preferences, API connections, and user access” but nothing happens when you click it.

It’s not clear what the different VCF files under the Pipeline Analysis are meant to illustrate. If they are just examples of what can be done then why are seven of them included? If they are meant to illustrate different analyses that can be performed then there should be an explanation for what each one is illustrating.

For some tools, when you first click on them nothing happens while the tool loads (e.g. MAF Tools). There should be some signifier that the tool is still loading.

The text on the Knowledge Base pages is too small to read, and not easy to make larger (on a Chrome browser).

Reviewer #2

(Remarks to the Author)

This manuscript presents OncoRisk, a web-based platform that integrates multiple oncology knowledge bases and pan-cancer cohorts to support variant-centric exploration, automated clinical reporting from VCF files, and cohort-level mutational analyses. The system is potentially useful for both research and clinical settings. However, there are still following comments:

Search/Query:

1. Regarding the Deep Query interface, the allowed query formats are described rather clearly in the Methods. Lines 157–158 state that “This function presents a detailed variant-focused page for any submitted variant identifier compatible with VEP (HGVS, RS ID, or one-line VCF),” which constrains the expected input types. However, lines 170–171 then use “BRAF V600E” as an example (“For BRAF V600E as an example, a total of 68 trials are returned, of which five are currently recruiting”). When I use “BRAF V600E” as the query term in the Deep Query interface on the live website, the system returns “No Results.” The authors should be more precise in how this example is presented, and explicitly document on the website which identifier formats are actually accepted by Deep Query, so that both users and readers can understand how to formulate valid queries.

In addition, there is an apparent inconsistency between modules: entering “BRAF V600E” in the Quick Search does return a result, whereas the same term fails in Deep Query. I suggest that the authors explicitly explain in the manuscript why the accepted query formats differ between Quick Search and Deep Query, and how users are expected to choose the appropriate identifier type for each module.

Pipeline Analysis:

2. Figure 3 is intended to illustrate “an automated workflow,” but panels 3A–3D mostly show interface snapshots and results, making it somewhat difficult to understand what each step of the pipeline is actually doing. I would suggest adding an additional subpanel before Figure 3A that schematically summarizes the workflow, and explicitly annotating how each step corresponds to the outputs shown in Figures 3A–3D. This would greatly help readers to map the conceptual pipeline to the concrete interfaces and results.

3. Figure 3A provides a clear view of the pipeline progress, which is very helpful. However, the blue status text at the bottom currently only shows “Running.” It would be more informative if this label indicated which specific step is being executed (e.g. annotation, filtering, report generation) on the live website, so that users can more clearly see which part of the workflow is in progress.

4. Variant annotation is performed using Ensembl VEP. Although the VEP Annotation section (around line 379) provides useful information about the software version, reference genome and the list of integrated annotation and prediction sources, it does not specify the actual VEP parameter settings. Ensembl VEP exposes a large number of configurable options, and some of them (especially filtering- and transcript-selection-related parameters) can substantially influence the final annotation output. Since the authors do not provide the underlying code or a detailed configuration (e.g. command-line options, plugin settings), it is difficult to assess the robustness and reliability of the reported results or to reproduce the pipeline locally.

In addition, it would be important to clarify whether the web server allows end users to adjust any key VEP parameters (e.g. choice of transcripts, filters, plugins), or whether the annotation runs with a fixed, non-configurable setup. If the latter, I would encourage the authors to state this explicitly and discuss the implications for flexibility and user-specific customization.

Database construction:

5. Lines 112–113: “We selected and obtained a total of 9 categories of oncogenic terms from CiVIC public API access and cached them in local storage.” Could the authors clarify why CiVIC was chosen as the sole source for defining these oncogenic term categories, rather than integrating similar concepts from other knowledge bases (e.g., OncoKB, ClinVar, CancerVar)? Is CiVIC considered sufficiently comprehensive for this purpose? A brief justification of this design choice would help readers understand the representativeness and potential limitations of the underlying term set.

6. For the database design, the robustness and reliability of a platform like OncoRisk critically depend on how different data

sources are integrated. The description in the Database Integration and API-Based Access section (around lines 328–343) provides only a broad overview. I would recommend that the authors add a summary table listing each data source included in OncoRisk (e.g., CiVIC, OncoKB, ClinVar, CancerVar, ClinicalTrials.gov, pan-cancer cohorts, etc.), together with basic statistics such as data volume, data types, and key attributes. In addition, an overview of the interconnections between these sources (for example, an entity-relationship diagram) would greatly help readers understand how the data are linked within OncoRisk.

Data usability:

7. In the Materials and methods section, line 340 state that “All processed data is made exportable in standard formats (CSV, JSON).” However, on the live website I was only able to find an “export” option in the knowledge base panel; I could not locate CSV/JSON export options for any of the analysis tool modules. It is therefore unclear whether I simply missed these controls, or whether the analysis tools currently do not support data export. In addition, when I clicked “export” in the knowledge base section, no download was triggered and no file was generated, which suggests that this functionality may not yet be fully implemented or may be affected by a bug. I recommend that the authors clarify in the manuscript which modules truly support CSV/JSON export, and ensure that the implemented behavior on the website matches the description.

8. It would be very helpful for users to be able to download the 3D objects and visualizations on the OncoRisk website. Could users export (i) the human anatomical model and protein structures shown in the Deep Query interface, and (ii) the network graphs generated in the Network Tools module (e.g., as images or graph files)?

Validation:

9. The tier-based variant classification system is a central component of OncoRisk and is described at a high level in the Results and Methods, with detailed rules relegated to Supplementary Table 2. While this is sufficient to make the procedure in principle reproducible, the current presentation does not convincingly establish that the scheme is either well-justified or empirically validated, which it is difficult to assess whether the chosen thresholds and weights yield clinically sensible behavior. I suggest to include a validation against existing standards or resources (e.g. comparing OncoRisk tiers to AMP/ASCO/ESMO-based classifications or to tools such as CancerVar on a curated set of actionable and benign variants).

10. Given that the authors position OncoRisk as a clinical decision-support tool for precision oncology and emphasize its use for patient-level reporting, I would have expected some evaluation in a real clinical workflow. For instance, one could compare OncoRisk-generated reports against conventional manual variant interpretation for a set of real patient cases, and assess concordance on key actionable variants, impact on treatment recommendations, and potential time savings.

11. The authors describe OncoRisk as “a state-of-the-art web server” in the title and, in the Discussion, provide a narrative comparison with several existing platforms. However, this comparison remains somewhat difficult to follow. I recommend adding a summary table that systematically contrasts OncoRisk with other tools (e.g., in terms of data sources, supported analyses, reporting capabilities, customizability), so that readers can more clearly appreciate the specific advantages and unique contributions of OncoRisk.

Others:

12. The manuscript currently does not include a Data Availability or Code Availability statement. To my understanding, Communications Biology generally requires authors to provide a clear description of where the underlying data and (when applicable) source code or analysis pipelines can be accessed, except in well-justified cases.

13. Website usability and layout: I have a few minor comments regarding the front-end design. (1) In the Server-Maftools and Pan-Cancer Visual modules, the main display panel on the right does not appear to adapt well to window resizing. On my screen, the content only occupies roughly two-thirds of the width, with the remaining one-third left as empty space, which reduces the effective use of the available area. (2) In Server-Maftools, within the “Data Source Selection” panel, the text inside the black box under “Upload Your MAF File” is truncated and not fully visible. Similarly, in the “File separator” dropdown, only “Tab” is fully visible, while the other options are cut off. I recommend that the authors improve the responsive layout and text rendering in these modules to enhance readability and user experience.

Reviewer #3

(Remarks to the Author)

I co-reviewed this manuscript with one of the reviewers who provided the listed reports. This is part of the Communications Biology initiative to facilitate training in peer review and to provide appropriate recognition for Early Career Researchers who co-review manuscripts.

Reviewer #4

(Remarks to the Author)

The manuscript presents OncoRisk, a user-friendly web server (<https://www.phenomeportal.org/oncorisk>) designed to facilitate cancer variant interpretation and analysis. The authors clearly describe the development of OncoRisk, including: (1) aggregation of variant annotations from multiple authoritative resources such as CiVIC, OncoKB, ClinVar, CancerVar,

ClinicalTrials.gov, and PDB; (2) the design of a clinical reporting pipeline with tier-based variant classification; (3) integration of seven curated, non-redundant pan-cancer cohorts to enable ready-to-use variant analysis; and (4) implementation of interactive visualization modules to support effective data exploration. Overall, the manuscript is well written and easy to follow.

Although several established platforms exist (e.g., cBioPortal and UCSC Xena), OncoRisk offers distinct advantages, particularly for cancer researchers with limited bioinformatics or programming expertise. The platform provides an intuitive interface, and each analytical module—Deep Query, Pipeline Analysis, Network Tools, MAF Tools, and Pan-Cancer Visualization—serves a clearly defined purpose and is straightforward to use. The breadth and quality of the visualization options are particularly noteworthy and facilitate interpretation of complex genomic data. Collectively, these features make OncoRisk a potentially valuable resource for the cancer research community.

That said, I have a few minor concerns that should be addressed prior to publication:

1. While OncoRisk offers a user-friendly interface, the documentation link (<https://docs.szapfs.org/oncorisk/>) is currently inaccessible. Effective utilization of any web-based analytical platform relies heavily on clear, comprehensive, and readily accessible documentation—especially when users are required to upload data in specific formats. The absence of functional tutorials or user manuals significantly limits usability and may hinder broader adoption of the resource.
2. Several figures included in the manuscript are of low resolution and remain difficult to interpret even when enlarged. I strongly recommend that the authors provide higher-quality images with adequate resolution to ensure readability and to allow proper technical evaluation.
3. Although the manuscript describes the web server implementation and briefly mentions limitations, additional details regarding technical and computational constraints would strengthen the work. Specifically, the authors should clarify:
 - a) the maximum dataset size supported for upload, analysis, and visualization;
 - b) how the platform handles incorrectly formatted or invalid input data; and
 - c) the number of concurrent users or simultaneous uploads that the system can reasonably support.

Addressing these points would improve transparency, reproducibility, and user confidence in the robustness and scalability of the OncoRisk platform

Version 1:

Reviewer comments:

Reviewer #1

(Remarks to the Author)

Dear editor,

Thank you for the opportunity to review the revised version of this manuscript. I have carefully examined the authors' responses and the updated manuscript, and I am satisfied that they have addressed all of my previous comments and questions thoroughly and appropriately. The revisions have improved the clarity and overall quality of the work. I have no further concerns and believe the manuscript is suitable for publication.

best,
DM

Reviewer #2

(Remarks to the Author)

The authors have adequately addressed my previous concerns and clarified the points I raised. I have no further major comments and recommend acceptance.

Reviewer #3

(Remarks to the Author)

I co-reviewed this manuscript with one of the reviewers who provided the listed reports. This is part of the Communications Biology initiative to facilitate training in peer review and to provide appropriate recognition for Early Career Researchers who co-review manuscripts.

Reviewer #4

(Remarks to the Author)

I thank the authors for their efforts in addressing all issues raised during the initial review. They have activated the link to the OncoRisk documentation, improved figure quality, and expanded the methodology section to include System Architecture and User Interface, Computational Performance and Scalability, and Data Robustness and Privacy. These revisions have substantially strengthened the manuscript. I therefore recommend the OncoRisk manuscript for publication.

Referee #1: Computational Genomics and Bioinformatics

Referee #2: Large scale consortia, Human Reference Atlas creation

Referee #3: Genomic Diagnostics & Bioinformatics

Reviewers' comments:

Reviewer #1 (Remarks to the Author):

Review of "OncoRisk: A state-of-the-art Web Server for bridging the oncogenic databases and pan-cancer cohorts to the translational oncology" by Song, et al.

The authors present OncoRisk, a server that integrates data from 10 oncogenic databases and seven pan-cancer cohorts, facilitating easy exploration of genomic variants, gene interactions, and therapy associations. It provides automated generation of a variety of tables, plots, and reports, enabling rapid assessment of various mutations. The server is very useful, enabling rapid exploration of a variety of cancer-related associations ranging from individual mutations to pan-cancer cohorts. While most, if not all, of the functions provided can be performed by other servers or software tools, it appears that OncoRisk uniquely integrates the specific combination of data sources and analysis functions available in a very user-friendly interface. It is likely to be a highly useful tool for the community.

We appreciate the reviewer's investigation into the hosting infrastructure of OncoRisk. We would like to clarify the relationships as follows:

1. My concerns have to do with the lack of transparency about the site it is hosted on. Phenomeportal.org appears to be owned by the Align Institute, a non-profit organization dedicated to building large, open biological datasets. However, this is not clearly explained on the phenomeportal.org website, nor is phenomeportal.org explicitly referenced on the Align website (alignbio.org).

The phenomeportal.org has **no formal or legal relationship with the Align Institute**. The domains phenomeportal.org and phenomeportal.com are owned and maintained purely by the research group of Professor Adil Mardinoglu at KTH Royal Institute of Technology, Sweden. OncoRisk is an independent academic project hosted within this portal. We have emphasized this point in the "Data and Code Availability": **The OncoRisk web server is freely accessible at <https://www.phenomeportal.org/oncorisk>.** (Lines 843)

Phenomeportal.org hosts the XOmics portal, which appears to be related to szaportal.com (which redirects to phenomeportal.org). The Documentation button on the main page directs to <https://docs.szapfs.org/oncorisk/>, which does not load. (I was unable to get the szaomics.com website to load.) Two of the manuscript authors are affiliated with SZA Omics R&D. Other tools on phenomeportal.org (e.g. GenRisk) appear to be publicly available, but the XOmics Portal requires an email to log in.

We clarify that OncoRisk is a strictly academic resource owned by Professor Adil Mardinoglu. While a collaboration with SZA Omics R&D (now renamed as Phenome Omics R&D) exists for providing the Demo tumor sequencing data (as disclosed in the Competing Interests policy), the server's long-term hosting and availability are managed by the academic institution (KTH and Scilifelab), independent of any commercial entity.

We have included the collaborations with Phenome Omics R&D in the "**Competing interests policy**" in the manuscript: "A.M. is a co-founder of SZA Longevity Inc. G.Y. and B.Y. are

employees of Phenome Omics R&D. The remaining authors declare no competing interests.”(Lines 835 to 836)

The XOmics portal and the legacy "szaportal" are **separate projects unrelated to OncoRisk**. We have now removed all redirects or links to these legacy or non-functional sites, ensuring a focused and reliable user experience.

Documentation: We apologize for the broken links. The documentation has been migrated and is now fully accessible on the main page of OncoRisk, via the “**Documentation**” and “**Help & Support**” buttons in “Help & Support”, which link to our document file: <https://xiyasong.github.io/OncoRisk/>. This document provides the general information, the help pages for analysis modules, and a technical contact email.

2. Furthermore, there is no explanation that I could find regarding the **privacy protocols for uploaded files**. Clearly a major beneficial feature of OncoRisk would be the ability to upload one’s own VCF or MAF files, but without clarity on who owns the website and what, if any, privacy rules apply to the uploaded data, most users will not be able or willing to use this functionality. Clearly the server is useful as-is for exploring public oncogenic databases. However, the lack of clarity about the relationships between the server website, the Align Institute, the SZA Omics corporation, and the manuscript authors raise questions about the ability to upload user data and the long-term availability of the server. I would like to see more clarity about the relationships between the authors, the Align Institute, and the SZA Omics corporation, as well as any pertinent commercial products related to OncoRisk, but I would not say such explanations are required for publication as the site is currently available and functional. More clarity about what happens to data that is uploaded, even if it is warnings that privacy cannot be guaranteed, would be highly desirable.

We take the reviewer's concern about user data seriously. We have now added a clear **privacy policy and data security clarification** at the bottom of the OncoRisk’s VCF data upload page, as the screenshot shows:

[figure redacted]

We also explicitly state the data privacy in the methods section of our manuscript under a new section named “**Data robustness and privacy**”: “To ensure maximum data privacy and security in pipeline analysis, all user-uploaded files are processed according to strict protocols. Each upload is anonymized using a Universally Unique Identifier (UUID), ... prevent

unauthorized data retention, all files are automatically purged from our secure server after 30 days.” (Lines 733 to 738).

3. Minor Issues: The buttons on the main server website are confusing. They react when you mouse over them, but nothing happens when you click on them. There seems to be different categories of buttons, however. Clearly the buttons in the Quick Actions panel are meant to do something when clicked, but none of them work. It’s not clear that anything is meant to happen when you click on the buttons in the Knowledge Base Overview or Latest Updates panels. If those are meant to just be static descriptors they should not change when you mouse over them.

We apologize for the confusion caused by the non-functional interactive elements. We have overhauled the user interface (UI) to ensure consistent behavior:

1. The "**Quick Actions**" panel buttons have been fixed and are now operational, directing users to the respective analysis modules: "Upload New Analysis" is for uploading new tumor sequencing data for running pipeline analysis; "Browse Knowledge Base" directs users to the first categorized terms, "Assertions", in our Knowledge Base. "Perform Deep Query" brings into the deep query interface, and "Documentation" brings into the formal OncoRisk document for help pages and general information. As the screenshot shows:

[figure redacted]

2. **Static** Descriptors: We have removed hover effects and animations from the static information panels (e.g., Knowledge Base Overview and Latest Updates) to distinguish them from clickable buttons.
4. The Help & Support button on the main page states that you can "Access documentation, tutorials, and contact technical support", but the button just takes you to the 7-minute video overview which is hosted on Google Drive (i.e. it's not on a publicly available YouTube channel).

As mentioned above, we have provided new text-based documentation fully accessible as a help page and hosted the video on the official YouTube channel @PhenomeLongevity: https://www.youtube.com/watch?v=q_cdku0P-cg. Once the user clicks the "Help" button in the upper-right corner, it will go directly to the YouTube video, which provides a quick overview of how to use the functions.

The information for contacting technical support has also been added to the main page of the OncoRisk document: hello@phenomelongevity.com.

5. The **Settings button** states that you can “Configure application preferences, API connections, and user access” but nothing happens when you click it.

We thank the reviewer for this critical investigation. The "Settings" button is now fully operational, allowing users to configure the UI themes and data refresh intervals. As the screenshot shows:

[figure redacted]

We have also added a dedicated **"API"** button and panel adjacent to the Settings for programming-access to the OncoRisk's knowledge base, compatible with Windows and Linux/Mac OS operating systems, with user examples provided. We have also introduced the API in the method section **"Database Integration and Exportation"** of our manuscript: "OncoRisk also provides a structured RESTful API that enables programmatic access to its integrated knowledge base across Windows, Linux, and macOS environments." (Lines 519-520)

6. It's not clear what the different VCF files under the Pipeline Analysis are meant to illustrate. If they are just examples of what can be done then why are seven of them included? If they are meant to illustrate different analyses that can be performed, then there should be an explanation for what each one is illustrating.

These samples are used as Demo files for the same process, which is our 'pipeline analysis' module, follows the same workflow as our new **Figure 4A** (see also attached figure in Reviewer #2's response). Only the patients' phenotypes are different (**Supplementary Table 1** for the patient metadata)

(part of new **Figure 4A**)

Seven of them (Note that only 6 unique samples were there, redundant upload of P00257_DNA_4 for showcase how the system works with redundant upload) were included because we tried to test the pipeline's running success rate by utilizing all Illumina somatic sequencing samples we have, as we received real patients' samples with different cancer diagnoses; we also want to show how they are analyzed and what the results are presenting. Also, all of them could serve as potential resources for the specific cancer type as case studies. In the future, we plan to analyze more cancer tissue samples using OncoRisk and provide it as an open resource.

We have described and revised the pipeline analysis's outputs for Demo samples in the manuscript: "To demonstrate the efficiency and utility of this functional module, we analyzed six tissue-derived DNA samples representing 5 different cancer types and one sample with endometriosis (**Supplementary Table 1**). The analysis results were shown as a demonstration on the web interface and in **Supplementary Figure 4**." (Lines 293-296)

Supplementary Figure 4. Pipeline-analysis results for 6 unique Demo somatic profiling samples. The top panel compares the total number of variants for each sample against key clinical and actionable subsets, including total variants, potential somatic actionable variants (somatic mutations with high-impact, oncogenic, or ClinVar pathogenic/likely pathogenic status), variants with CIViC evidence, and the sum of all ClinVar pathogenic, likely pathogenic, and conflicting variants. The bottom panel shows the composition of ClinVar variants for each sample.

Then, we have enhanced the results section by including specific sample IDs for the demo cases, providing greater detail and ensuring a clearer presentation of our findings: "The sample with endometrial cancer(with ID P00257_DNA_1) is a rare type of cervical adenocarcinoma. The pipeline results contain... that may support the development of new therapeutic strategies." (Lines 298-317).

7. For some tools, when you first click on them nothing happens while the tool loads (e.g. MAF Tools). There should be some signifier that the tool is still loading.

We have added "Initializing" loading pages for these two modules. Server-Maftools and Pan-cancer-visual are two separate modules built based on docker image and on Sciliflab Serve. It takes less than 30 seconds to initiate. As like what we tested, the first time a user clicks the module will take a bit longer than even after the loading page finishes, but from the second time, the entrance will be quite fast and connect smoothly after the loading page finishes. As the screenshot shows:

8. The text on the Knowledge Base pages is too small to read, and not easy to make larger (on a Chrome browser).

We have now significantly increased the base **font size** by **50%** across all Knowledge Base content to ensure the text is much clearer and legible. It has improved the readability on the website.

As the screenshot shows:

Name	Aliases	Variants	Assertions	Associated Diseases	Associated Evidence
BRAF V600E	RS113488022 VAL600GLU +2	V600E	AID12 AID20 +3	Colorectal Ca. Skin Melanoma +35	EID20 EID12 +3
ERBB2 Amplification	OVEREXPRESSION	Amplification	AID32 AID1 +1	Breast Cance Gastroesophag +24	EID32 EID1 +1
BRC1A1 Mutation		Mutation		Ovarian Cancer Castration-re +9	
v-ALK Fusion	REARRANGEMENT ALK Fusion	Fusion	AID3 AID34 +3	Lung Non-smal Anaplastic La +10	EID3 EID34 +3
KRAS Mutation		Mutation		Colorectal Ca Lung Non-smal +12	
NPM1 EXON 11 MUTAT...	EXON 12 MUTAT...	EXON 11 MUTAT...		Acute Myeloid...	
DNMT3A R882	R693 ARG882 +1	R882		Acute Myeloid...	

Reviewer #2 (Remarks to the Author):

This manuscript presents OncoRisk, a web-based platform that integrates multiple oncology knowledge bases and pan-cancer cohorts to support variant-centric exploration, automated clinical reporting from VCF files, and cohort-level mutational analyses. The system is potentially useful for both research and clinical settings. However, there are still following comments:

Search/Query:

1. Regarding the Deep Query interface, the allowed query formats are described rather clearly in the Methods. Lines 157–158 state that “This function presents a detailed variant-focused page for any submitted variant identifier compatible with VEP (HGVS, RS ID, or one-line VCF),” which constrains the expected input types. However, lines 170–171 then use “BRAF V600E” as an example (“For BRAF V600E as an example, a total of 68 trials are returned, of which five are currently recruiting”). When I use “BRAF V600E” as the query term in the Deep Query interface on the live website, the system returns “No Results.” The authors should be more precise in how this example is presented, and explicitly document on the website which

identifier formats are actually accepted by Deep Query, so that both users and readers can understand how to formulate valid queries.

We appreciate this point; we have provided detailed guidelines for the data input format on the website, in the documentation, in the video, and in the corresponding manuscript.

We have now also added an **"Example search format"** that clearly defines the expected input types for Deep Query on the webpage and the corresponding manuscript. As the screenshot shows:

For the manuscript: When we say we search BRAF V600E, we are using the relevant rsID for BRAF V600E (rs113488022) retrieved from a quick query, as a search term for a deep query. We have modified the description for Quick Search: "This function also supports searches by variant nomenclature, including rsID; for example, querying "rs1134880" returns the BRAF V600E profile, and vice versa, searching BRAF V600E retrieves its rsID in the corresponding detail page." (Lines 140 to 143) and for Deep Query: "Using rs113488022 (the rsID for BRAF V600E) as a query example, a total of 68 trials are returned, of which five are currently recruiting..." (Lines 214-216)

In addition, there is an apparent inconsistency between modules: entering "BRAF V600E" in the Quick Search does return a result, whereas the same term fails in Deep Query. I suggest that the authors explicitly explain in the manuscript why the accepted query formats differ between Quick Search and Deep Query, and how users are expected to choose the appropriate identifier type for each module.

The **Quick Search** is based on our knowledge base built using fuzzy matching for any of CiViC oncogenic terms, not only variant search but also drug, gene, drug, and disease-centric search; for variant-centric, it provides a fast approach for connecting oncogenic terms to a query variant and simplifies searching for variant synonyms (e.g., searching "BRAF V600E" to retrieve its RSID).

To expand query possibilities to all potential variants across the genome, we designed **Deep Query**, an Ensembl VEP-based system focused on processing any of the genomic mutations (currently on SNVs). Deep Query performs VEP annotation via customized command lines across multiple databases (CiViC, ClinVar, etc.) and integrates the oncogenic family tree, 3D protein structures, and clinical trials for deep-level investigation.

Consequently, Deep Query requires a strict input format to ensure VEP runs correctly. We recommend using HGVS notation with RefSeq transcripts (NM) or RSID, though other VEP-compatible formats, such as one-line VCF, are also supported.

In summary, Quick Search and Deep Query complement each other to ensure the platform functions effectively based on different aspects that the user intends to achieve.

Regarding these points, we have firstly changed the “Variant Query” module in **Figure 1** to explicitly show the quick search and deep query module on our web server overview (inside the yellow box):

[figure redacted]

To assist users in choosing the correct module, we have added guidelines and example inputs to both our webpage and documentation.

We have also revised the old Figure 2 into the new **Figure 2** and **Figure 3** to improve the readability (Reviewer #4's suggestion) and independence of the Deep Query module.

Figure 2 is now for knowledgebase & Quick Search & Network Tools:

[figure redacted]

Figure 2. Integrated Variant Query Interfaces: Knowledge Base, Quick Search and Network Visualization

With the text revised accordingly:" The first query function, 'Quick Search" box, is designed for fuzzy matching any of a single oncogenic term of interest across all term categories, including gene names, fusions, or variants formatted with amino acid alteration (**Figure 2B**). "(Lines 138-140)

And **Figure 3** for Deep Query:

And as text description modified as follows: “We developed a submission-driven Deep-Query interface serving as a complementary function for analyzing any genomic variants, including variants that are absent from CiVIC (**Figure 3A**), such as ... format.”(Lines 194-199)

Pipeline Analysis:

2. Figure 3 is intended to illustrate “an automated workflow,” but panels 3A–3D mostly show interface snapshots and results, making it somewhat difficult to understand what each step of the pipeline is actually doing. I would suggest **adding an additional subpanel before Figure 3A that schematically summarizes the workflow, and explicitly annotating how each step corresponds to the outputs shown in Figures 3A–3D**. This would greatly help readers to map the conceptual pipeline to the concrete interfaces and results.

We appreciate this suggestion. We have revised our old Figure 3 into new **Figure 4** for the pipeline analysis module, adding a workflow panel as the first sub-figure. Since all of our snapshots reveal the user interface and PDF report, we highlighted this in a box on the left side of **Figure 4B**. We also revised the corresponding manuscript to make the process clearer to readers:" The workflow is shown in **Figure 4A**. The system allows direct upload of individual sequencing VCF files containing thousands of somatic mutations. The liftover (GRCh37 to GRCh38) is performed if the input file is in the GRCh37 assembly. After a quality filter to only retain variants with the "PASS" flag,...CancerVar Tiers, etc" (Lines 246 to 283).

[figure redacted]

Figure 4. Rapid, individualized tumor profiling and reporting via an automated workflow.

We also uploaded the source code for the analysis to the GitHub repository (https://github.com/xiyasong/OncoRisk_Codes/blob/main/pipelines/oncology_vep_liftover_pass_anno.sh).

3. Figure 3A provides a clear view of the pipeline progress, which is very helpful. However, the blue status text at the bottom currently only shows "Running." It would be more informative if this label indicated which specific step is being executed (e.g. annotation, filtering, report generation) on the live website, so that users can more clearly see which part of the workflow is in progress.

We appreciate this suggestion. Currently, a somatic profile (fewer than 5,000 variants) typically takes only a few seconds per sample. The pipeline sequentially performs filtering to retain only 'PASS' variants, liftover to GRCh38 of the detected input of GRCh37 assembly, VEP annotation with JSON output, and finally, conversion to the front-end UI and PDF report. Except for the VEP annotation process, all other process is nearly instant processes.

To address the reviewer's point, we have updated the process indicator to reveal the VEP process as shown in the screenshot below. However, since most of these processes are nearly instantaneous, it may appear very rapid to the user.

4. Variant annotation is performed using Ensembl VEP. Although the VEP Annotation section (around line 379) provides useful information about the software version, reference genome and the list of integrated annotation and prediction sources, it does not specify the actual VEP parameter settings. Ensembl VEP exposes a large number of configurable options, and some of them (especially filtering- and transcript-selection-related parameters) can substantially influence the final annotation output. Since the authors do not provide the underlying code or a detailed configuration (e.g. command-line options, plugin settings), it is difficult to assess the robustness and reliability of the reported results or to reproduce the pipeline locally.

As previously mentioned, we have provided the VEP command to the [GitHub repository](https://github.com/xiyasong/OncoRisk_Codes/blob/main/pipelines/oncology_vep_liftover_pass_anno.sh) (https://github.com/xiyasong/OncoRisk_Codes/blob/main/pipelines/oncology_vep_liftover_pass_anno.sh) and revised the "VEP Annotation" section to provide more details.

We also added more details in the manuscript's method section under VEP Annotation: " We utilize the --pick flag with a defined order: '--pick_order mane_select, rank' to prioritize MANE Select transcript when multiple RefSeq isoforms are available for a single variant. The annotation also incorporates..."(Lines 590-593)

In addition, it would be important to clarify whether the web server allows end users to adjust any key VEP parameters (e.g. choice of transcripts, filters, plugins) or whether the annotation runs with a fixed, non-configurable setup. If the latter, I would encourage the authors to state this explicitly and discuss the implications for flexibility and user-specific customization.

We appreciate the reviewer's inquiry regarding parameter customization. Since somatic VCF files are typically small, the system is designed to perform VEP annotation on all variants without the need for aggressive pre-filtering. To ensure the stability of downstream analysis and clinical PDF report generation, we used a fixed configuration with VEP setting to RefSeq transcripts, which are the standard for clinical requirements. All relevant functional plugins are enabled by default, as the small file sizes allow for comprehensive annotation without compromising performance. We also provided full VEP and deep-queried outputs in JSON format for the user to download, and it could be helpful for the user to perform any flexible investigation.

We have added a statement in the discussion as well: "The VEP annotation runs with a fixed setup, which future work may aim to add more flexibility to this process."(Lines 823-824)

In addition, taken this suggestion, we implement a new "**post-filter**" function that allows users to perform post-filtering based on some key VEP annotation results in JSON files, as we described in the manuscript: " Lastly, in the user interface, there is also a 'Filters' function for user directly apply post-filtering based on some key VEP annotation results in JSON files, including chromosomes, variant consequences, OncoKB annotation, CancerVar Tiers, etc."(Lines 290-292).

The post-VEP-filter is as the screenshot shows:

Database construction:

5. Lines 112–113: “We selected and obtained a total of 9 categories of oncogenic terms from CiViC public API access and cached them in local storage.” Could the authors clarify why CiViC was chosen as the sole source for defining these oncogenic term categories, rather than integrating similar concepts from other knowledge bases (e.g., OncoKB, ClinVar, CancerVar)? Is CiViC considered sufficiently comprehensive for this purpose? **A brief justification** of this design choice would help readers understand the representativeness and potential limitations of the underlying term set.

We selected CiViC as our primary source for defining oncogenic categories because it offers a fully open-sourced, frequently updating, highly structured, categorized, community-vetted ontology terms database, which extends beyond simple pathogenicity.

In oncology research, we noticed that a significant challenge exists in bridging clinical terms and druggable biomarkers, such as “ALK fusions,” “EGFR exon 19 mutations,” or biomarkers like TMB and MSI, with the single genomic variations data found in sequencing files, like RSIDs and genomic location coordinates. While ClinVar and CancerVar focus primarily on single-variant pathogenicity (ACMG/AMP scale) and OncoKB emphasizes therapeutic actionability (Levels 1–4) for a group of mutations, as we mentioned above (ALK fusions, EGFR exon 19 mutations, which are a form of gene-specific mutation types), often without direct genomic mapping, CiViC effectively “flattens” these complex groups by creating a new way of structuring data into **Molecular Profiles** (functional groups), **Variants** (individual mutations), and **Features** (genes, fusions, or factors like TMB), etc(https://docs.civicdb.org/en/latest/model/molecular_profiles.html#molecular-profiles). In summary, we think that CiViC provides a clear framework to connect sequencing data with clinical nomenclature and also to connect all terms, which form the basis for our network-based query system. This multidimensional oncogenic terms database allows OncoRisk to classify evidence across multidimensional aspects, and then we complement it with specific pathogenicity and druggability data from ClinVar, OncoKB, and CancerVar.

We have added a brief justification for this in our manuscript under the **method** section **Database Integration and Exportation**: “Specifically, we utilized CiViC as the foundational ontology for defining oncogenic categories and constructing the knowledge base. We chose this resource due to its open-source, highly structured nature and its unique capacity to map diverse clinical biomarker nomenclatures (e.g., variant groups and fusions) to specific genomic alterations. Then, we complemented it with specific pathogenicity and druggability data from other databases. These data are...” (Lines 501 to 506)

We have also made a brief justification in the **results** section clearer regarding this point: “The OncoRisk knowledge Base was designed and built based on CiVIC for its fully open-source, highly structured, and massive community-based knowledge records. Then, other databases (OncoKB, ClinVar, CancerVar) were complemented to provide comprehensive integrated information. Firstly, we...”(Linex 119 to 122)

6. For the database design, the robustness and reliability of a platform like OncoRisk critically depend on how different data sources are integrated. The description in the Database Integration and API-Based Access section (around lines 328–343) provides only a broad overview. I would recommend that the authors add a **summary table** listing each data source included in OncoRisk (e.g., CiVIC, OncoKB, ClinVar, CancerVar, ClinicalTrials.gov, pan-cancer cohorts, etc.), together with basic statistics such as data volume, data types, and key attributes. In addition, an overview of the interconnections between these sources (for example, an **entity-relationship diagram**) would greatly help readers understand how the data are linked within OncoRisk.

We have now added a summary table for all resources we integrated (**Supplementary Table 2**) that summarizes the data volume, data types, and key attributes. We also made an overview, an entity-relationship diagram, as **Supplementary Figure 5**. As we fixed in the manuscript, **Methods** part "Database Integration and Exportation": "OncoRisk has integrated multiple resources to achieve comprehensive functions in all modules, as summarized in the **Supplementary Table 2** and **Supplementary Figure 5**." (Lines 498-499)

Module	Data Source	Data Volume	Data Types & Key Attributes	Integration Role
Knowledge Base & Quick Search; Deep Query & Pipeline Analysis	CiVIC	Pre-cached Oncogenic terms of 9 main categories from API access; Downloaded VCF (20250630)	Assertions, evidence, molecular profiles, features, variants, disease, therapies, phenotypes, sources	Defines the backbone for oncogenic terms and variant mapping for main knowledge base interface
	OncoKB	Real-time API / caching	Oncogenicity, Mutation Effect, Therapeutic Evidences, FDA Level	Provides precision oncology therapeutic tiering and drug actionability
	ClinVar	Read-time API / caching & Downloaded VCF (2025 ClinVar (20250623))	Variant Germline Pathogenicity and Somatic Clinical impact (Tiers & Review status)	Validates clinical significance and complementing germline/somatic classification
Deep Query & Pipeline Analysis	CancerVar	Real-time API / caching	Oncogenicity scoring, Automated AMP/ASCO interpretation results	Provides AI-empowered automated scoring for mutations
	ClinicalTrials.gov	Real-time API / caching	Clinical trial status (Recruiting, Active), Phases, Study details	Links actionable variants to currently available clinical trials.
Deep Query	Disease Ontology / OncoTree	Full databases via downloaded files	Standardized disease nomenclature & disease hierarchy	Enables hierarchical disease queries and maps specific subtypes to broader cancer families and tissue-level grouping.
	RCSB PDB	Full databases via downloaded files	3D Protein-Ligand-mutant complex structures	Visualizes mutation locations relative to drug binding pockets.
Pan-Cancer Visual	AACR Project GENIE (v18.0)	250,018 samples	Somatic mutations profiles via different sequencing technologies, ethnicities, cancer types and coverage	For querying sepecific variant in the cohort-somatic mutation profile & Pre-calculated Variant Allele Frequency (VAF) and real-world mutation prevalence and gene-based mutation prevalence
	MSK-CHORD	25,040 samples		
	China Pan-Cancer PCAWG	10,194 samples		
	Other Cohorts (MSC, MSS, SUMMIT)	2,922 samples		
Server-Maftools	TCGA & CCLE	33 TCGA Cohorts; 2,427 Cell Lines (DepMap 2024 Q2)	MAF files with Clinical metadata	Used in "Server-Maftools" for all analys and plotting

Supplementary Table 2. Overview of integrated database & cohort volume, type, and key attributes

Supplementary Figure 4

Lastly, we also showed the data size on the Server-Maftools webpage and the Pan-Cancer-Visual webpage. See the screenshots:
 Server-Maftools:

Available TCGA Cohorts

The following TCGA cohorts are available for analysis:

Search:

Study_Abbreviation	source	n_samples
ACC	MC3 Firehose CCLE	92 62 NA
BLCA	MC3 Firehose CCLE	411 395 NA
BRCA	MC3 Firehose CCLE	1026 978 NA
CESC	MC3 Firehose CCLE	291 194 NA
CHOL	MC3 Firehose CCLE	36 35 NA
COAD	MC3 Firehose CCLE	406 367 NA
DLBC	MC3 Firehose CCLE	37 48 NA
ESCA	MC3 Firehose CCLE	185 185 NA
GBM	MC3 Firehose CCLE	400 283 NA
HNSC	MC3 Firehose CCLE	509 511 NA
KICH	MC3 Firehose CCLE	66 66 NA
KIRC	MC3 Firehose CCLE	370 476 NA
KIRP	MC3 Firehose CCLE	282 282 NA
LAML	MC3 Firehose CCLE	140 193 NA
LGG	MC3 Firehose CCLE	525 516 NA

Showing 1 to 15 of 34 entries

Previous **1** 2 3 Next

Pan-Cancer-Visual:

Data Table of Cohort Metrics

Show 10 entries

Search:

Summary statistics for all loaded cohorts.

Cohort	Sample Size	Total Mutations	Cancer Types	Mean Mutations per Sample	Median Mutations per Sample
GENIE (n=250018)	250018	2738934	112	13.41	5
MSK-CHORD (n=25040)	25040	208953	5	8.75	5
China Pan-Carcinoma (n=10194)	10194	99895	25	10.18	6
PCAWG (n=2922)	2922	382937	30	142.73	59
Metastatic Solid Carcinomas (n=500)	500	87070	43	174.14	67
MSS Mixed Solid Tumors (n=249)	249	167307	7	671.92	345
SUMMIT (n=141)	141	1042	22	9.83	6.5

Showing 1 to 7 of 7 entries

Previous **1** Next

Data usability:

7. In the Materials and methods section, line 340 state thaty “All processed data is made exportable in standard formats (CSV, JSON).” However, on the live website I was only able to find an “export” option in the knowledge base panel; I could not locate CSV/JSON export options for any of the analysis tool modules. It is therefore unclear whether I simply missed these controls, or whether the analysis tools currently do not support data export.

In addition, when I clicked “export” in the knowledge base section, no download was triggered and no file was generated, which suggests that this functionality may not yet be fully implemented or may be affected by a bug. I recommend that the authors clarify in the manuscript which modules truly support CSV/JSON export, and ensure that the implemented behavior on the website matches the description.

We appreciate that the reviewer mentioned the inaccessibility of the previous exporting options. We have addressed this by adding clear and functioning **"Export Data"** buttons across all analysis modules to match the manuscript description. Users can now download data in **JSON** formats from all Knowledge Base detail pages (including Assertions, Evidence, and Molecular Profiles), Deep Query results, and the Pipeline Output for a single VCF file.

We have updated the content of the manuscript regarding this new feature in the methods **"Database Integration and Exportation"**: "All processed data (Oncogenic terms in Knowledge Bases, Deep Query results, and pipeline analysis outputs) is made exportable in JSON formats. This also ... RESTful API that enables programmatic access to its integrated knowledge base across Windows, Linux, and macOS environments." (Lines 516-520) And also highlighted in the results section, "The queried results from quick search and deep query can be exported via downloadable JSON outputs," (Lines 99-100) and "through a dedicated interface that allows manual adjustment and exportation of organized outputs (JSON and PDF)," (Lines 103-104)

8. It would be very helpful for users to be able to download the 3D objects and visualizations on the OncoRisk website. Could users export (i) the human anatomical model and protein structures shown in the Deep Query interface, and (ii) the network graphs generated in the Network Tools module(e.g., as images or graph files)?

Regarding the export of visualizations, we have implemented high-resolution "Snapshot" buttons for both the 3D protein structures, anatomical models, and networks. We believe providing rendered image captures is more practical for most researchers than exporting raw 3D mesh files and raw graph files, which require specialized software to view.

Nevertheless, in the protein structure page, we clearly provided the **PDB ID** of this complex, which should be able to lead the user to access the RCSB PDB portal (<https://www.rcsb.org/>) and directly download it directly.

Please see the newly added Snapshot button:

Drug	Vemurafenib
Cancer Type	Erdheim-Chester Disease
Source	OncoKB
Tier	FDA Approved
Classification	TRANSFERASE/TRANSFERASE INHIBITOR
Protein Chains	2
Small Molecules	1
Mutations	1
Resolution	2.99 Å

In addition, since we provided the **source code** for 3D protein visualization (https://github.com/xiyasong/OncoRisk_Codes/tree/main/3D-Mutant-Drug) and network graph generation (https://github.com/xiyasong/OncoRisk_Codes/tree/main/Disease_Hierarchy), we believe this enhanced accessibility for these two modules. Running the source code with the input as the same as the query in the OncoRisk webpage could directly generate the HTML-formatted cancer family tree we built.

Validation:

9. The tier-based variant classification system is a central component of OncoRisk and is described at a high level in the Results and Methods, with detailed rules relegated to **Supplementary Table 2**. While this is sufficient to make the procedure in principle reproducible, the current presentation does not convincingly establish that the scheme is either well-justified or empirically validated, which it is difficult to assess whether the chosen thresholds and weights yield clinically sensible behavior. I suggest to **include a validation against existing standards or resources (e.g. comparing OncoRisk tiers to AMP/ASCO/ESMO-based classifications or to tools such as CancerVar on a curated set of actionable and benign variants)**.

OncoRisk's Tier design is not intended to develop a de novo scheme for variant classification; rather, we want it to serve as an integrated framework that synthesizes data from established authorities, including OncoKB, CiViC, CancerVar, and ClinVar, so that the resources complement each other when any resource lacks valid data. For example, ClinVar's "SCI" field (somatic integrated clinical impact) was recently developed (from mid 2025), so there isn't much information there, but variants still get clinical impact scores from OncoKB and CiViC.

Based on the reviewer's suggestion, we first reviewed our **OncoRisk tiering** rules and made the following changes:

(1) We have included the ClinVar's aggregate somatic clinical impact for this single variant from the "SCI" field and aggregate oncogenicity classification from the "ONC" field for the variant in the ClinVar annotation file. Based on ClinVar, the SCI is based on the recommendations from AMP/ASCO/CAP, and the ONC is based on the recommendations from ClinGen/CGC/VICC.

(2) Together with previously included "OncoKB classifies as 'Oncogenic'", ClinVar classifies as 'Pathogenic' with some minor score adjustment using CancerVar and AlphaMissense (See new tier rules in **Supplementary Data 1**), we defined the "clinical significance" as the main score follows the well-known and authoritative judgment for the variant.

Any of these that bring a clinical significance of 5 will immediately bring a variant to **Tier 1**.

Condition	Score	Notes
BASE SCORE		
OncoKB classifies as "Oncogenic"	5	
ClinVar classifies as "Pathogenic"	5	
AMP Somatic Tier I (any strength)	5	Strong/Potential clinical significance
ClinVar ONC = "Oncogenic"	5	ClinVar oncogenicity field
OncoKB classifies as "Likely Oncogenic"	4	
ClinVar classifies as "Likely Pathogenic"	4	
ClinVar classifies as "Drug Response"	3	Pharmacogenomic relevance
ClinVar "Uncertain" or OncoKB "Unknown"	1	Variant of uncertain significance
No classification available	0	
ADJUSTMENTS		
		Applied after base score
CancerVar supports pathogenicity	+0.5	Only if score < 5
AlphaMissense predicts pathogenic	+0.5	Only if score < 5
ClinVar has conflicting interpretations	-1.0	Uncertainty penalty
AlphaMissense contradicts (predicts benign)	-0.5	Only if already pathogenic
CancerVar contradicts (Tier IV / Benign)	-0.5	Only if already pathogenic
AMP = "Tier_IV_-, Benign/Likely B"	-1	Benign/Likely benign penalty
Final score clamped to range 0-5		

Part of (new **Supplementary Data 1**)

And now we can confirm that, instead of redefining clinical impact, OncoRisk is simply adding more dimensions for showing variant information, such as prognostic effects, ongoing clinical trials, and population rarity, into a unified, quantifiable module, where in the radar chart, the user can explore a variant on all aspects.

Using this new tiering table, we have performed a validation to see how this tier assigns a curated set of variants that follows the reviewer's suggestion:

A reference dataset was derived from the ClinVar VCFs that were previously used in VEP analysis (version 20250623) using Python (cyvcf2 and pandas). Variants were initially filtered

based on the presence of ONC (oncogenicity) or SCI (somatic clinical impact) annotations in the INFO field. From this subset, variants were stratified into two control groups based on clinical significance: Positive Control Group: Included variants annotated as "Oncogenic" (randomly sampled, n=20) or "Tier_I_-_Strong" (all available records,n=5). Negative Control Group: Included all variants annotated as "Benign" or "Likely_benign" under either the ONC or SCI (Tier IV) classification systems (n=16).

Then, we perform the deep query on each of them to get their OncoRisk score assessment and their corresponding attributes. Results are shown in the attached figure and as a Supplementary Figure for the OncoRisk tier VS ClinVar ONC value.

Supplementary Figure 3. Validation of OncoRisk Final Tier and scores compared to the original ONC values from ClinVar. The majority of oncogenic mutations from the positive control group are correctly prioritized as Tier 1, with their final quantitative scores reflecting the depth of integrated evidence. The assignment of certain variants to Tier 2 demonstrates the system's refined evidence-weighting adjustments. Conversely, most benign and likely benign variants from the negative control group are classified as Tier 3.

We can see that most oncogenic mutations were assigned to Tier 1, with their final scores representing the depth of collected evidence. Some variants shifted to Tier 2 based on specific scoring adjustments, while most benign or likely benign variants were classified as Tier 3. A detailed comparison table can be found in **Supplementary Data 2**.

We have added this plot in the manuscript: " Variants that contain high importance assigned by OncoKB oncogenicity or ClinVar pathogenicity, Oncogenicity (ONC), and AMP tier (SCI field) will be assigned as Tier 1. A detailed validation of OncoRisk Tiers is attached in **Supplementary Figure 3 and Supplementary Data 2.**" (Lines 209-212)

Lastly, we added a **limitation** to declare the OncoRisk's Tier: " The Tier and scores calculated by OncoRisk represent the information that can be gathered; it is not intended to replace the standard AMP/ASCO interpretation guidelines." (Lines 821-823)

10. Given that the authors position OncoRisk as a clinical decision-support tool for precision oncology and emphasize its use for patient-level reporting, I would have expected some evaluation in a real clinical workflow. For instance, one could compare OncoRisk-generated reports against conventional manual variant interpretation for a set of real patient cases, and assess concordance on key actionable variants, impact on treatment recommendations, and potential time savings.

We appreciate the reviewer's perspective on clinical evaluation. The core strength of OncoRisk lies in all four functional modules, not only the pipeline-analysis module, which, together, aim to enhance the efficiency of initial data collection for both researchers and clinicians.

While OncoRisk's "pipeline analysis" module is designed with a PDF output to assist in clinical decision-support, it is currently positioned as a web-tool helping the process rather than a platform for direct clinical diagnostics. We do not yet have formal collaborations with clinical oncology boards for real-time reporting. However, we have demonstrated OncoRisk's utility by multiple querying examples and analyzed the uploaded real patient samples as case demos to show how the system identifies potential oncogenic mutations and provides exportable results. We have now added a **Limitation** section in the **Discussion** to explicitly state that: "While OncoRisk's pipeline analysis provides a quick approach for initial evidence synthesis on clinical usage, it is currently positioned as a web tool helping the process rather than a fully practical diagnostic system."(Lines 819-821)

11. The authors describe OncoRisk as "a state-of-the-art web server" in the title and, in the **Discussion**, provide a narrative comparison with several existing platforms. However, this **comparison** remains somewhat difficult to follow. I recommend **adding a summary table** that systematically contrasts OncoRisk with other tools (e.g., in terms of data sources, supported analyses, reporting capabilities, customizability), so that readers can more clearly appreciate the specific advantages and unique contributions of OncoRisk.

We appreciate the reviewer's recommendation. To address this, we have added a comparison table (**Supplementary Table 3**) in the Discussion section. This table systematically contrasts OncoRisk with the web servers we previously mentioned in the discussion section (e.g., cBioPortal, PANDA, Onkopos), and we have expanded the comparison to include four major foundational knowledge bases and tools: OncoKB, CIViC, COSMIC, and CGI(cancer genome interpreter), etc. As we added in the **discussion** section of the manuscript: "In summary, we systematically compared OncoRisk with over 10 other databases/servers to emphasize the features of OncoRisk (**Supplementary Table 3**)."(Lines 812-814)

Others:

12. The manuscript currently does not include a Data Availability or Code Availability statement. To my understanding, Communications Biology generally requires authors to provide a clear description of where the underlying data and (when applicable) source code or analysis pipelines can be accessed, except in well-justified cases.

We have now added a **Data and Code Availability** statement at the end of the manuscript:

"The OncoRisk web server is freely accessible at <https://www.phenomeportal.org/oncorisk>. The pan-cancer cohort datasets analyzed in this study are publicly available via cBioPortal,

with the exception of the AACR Project GENIE dataset (v18.0), which was accessed via Synapse under application ID ACT-2119. TCGA and CCLE data were retrieved using the TCGAmutations and maftools R packages. Clinical knowledge data (including CiVIC, ClinVar, and CancerVar) were accessed via their public APIs or bulk VCF downloads, while OncoKB data were accessed via API under an approved academic license. All other data supporting the findings of this study are available within the article and its Supplementary Information.

The source code for the OncoRisk web server is available at the public GitHub Repository: https://github.com/xiyasong/OncoRisk_Codes. (Lines 843-852)

13. Website usability and layout: I have a few minor comments regarding the front-end design. (1) In the Server-Maftools and Pan-Cancer Visual modules, the main display panel on the right does not appear to adapt well to window resizing. **On my screen, the content only occupies** roughly two-thirds of the width, with the remaining one-third left as empty space, which reduces the effective use of the available area. (2) In Server-Maftools, within the “Data Source Selection” panel, the text inside the black box under **“Upload Your MAF File”** is truncated and not fully visible. Similarly, in the “File separator” dropdown, only “Tab” is fully visible, while the other options are cut off. I recommend that the authors improve the responsive layout and text rendering in these modules to enhance readability and user experience.

In both **Server-maftools** and **Pan-cancer-visual**, we have made the boxes **fill the window** and ensured that the figures automatically fit the browser's window size. The problem of text not being fully visible was fixed by **correcting** the CSS of the text window within the “Data source selection” panel; additionally, **radio buttons** are now used for the file separator instead of a dropdown menu. The font size of **several** boxes has been enlarged. In **Server-maftools**, we have now added a tool-specific updates section to track regular improvements. Please see the following screenshots:

Data Source Selection

Choose your data source:

Load TCGA/CCLC Cohort Data Upload My Own MAF File

Upload Your MAF File

Browse... ACC-TP.final_analysis_set.maf

Upload complete

Upload your MAF file

Supported formats: .maf, .txt, .tsv

Maximum file size: 100 MB

File has header row

File separator:

Tab

Comma

Uploaded Data Info

Basic Statistics:

Total Samples: 62
 Total Variants: 9540
 Total Genes: 5845
 File: ACC-TP.final_analysis_set.maf

Column Information:

Total Columns: 339
 VAF-related: 1
 VAF Columns: i_dbSNPPopFreq

All Column Names:

Hugo_Symbol, Entrez_Gene_Id, Center, NCBI_Build, Chromosome, Start_Position

=====
 =====
 Reviewer #3 (Remarks to the Author):

I co-reviewed this manuscript with one of the reviewers who provided the listed reports. This is part of the Communications Biology initiative to facilitate training in peer review and to provide appropriate recognition for Early Career Researchers who co-review manuscripts.

=====
 =====
 Reviewer #4 (Remarks to the Author):

The manuscript presents OncoRisk, a user-friendly web server (<https://www.phenomeportal.org/oncorisk>) designed to facilitate cancer variant interpretation and analysis. The authors clearly describe the development of OncoRisk, including: (1) aggregation of variant annotations from multiple authoritative resources such as CiViC, OncoKB, ClinVar, CancerVar, ClinicalTrials.gov, and PDB; (2) the design of a clinical reporting pipeline with tier-based variant classification; (3) integration of seven curated, non-redundant pan-cancer cohorts to enable ready-to-use variant analysis; and (4) implementation of interactive visualization modules to support effective data exploration. Overall, the manuscript is well written and easy to follow. Although several established platforms exist (e.g., cBioPortal and UCSC Xena), OncoRisk offers distinct advantages, particularly for cancer researchers with limited bioinformatics or programming expertise. The platform provides an intuitive interface, and each analytical module—Deep Query, Pipeline Analysis, Network Tools, MAF Tools, and Pan-Cancer Visualization—serves a clearly defined purpose and is straightforward to use. The breadth and quality of the visualization options are particularly noteworthy and facilitate the interpretation of complex genomic data. Collectively, these features make OncoRisk a potentially valuable resource for the cancer research community.

That said, I have a few minor concerns that should be addressed prior to publication:

1. While OncoRisk offers a user-friendly interface, the documentation link

(<https://docs.szapfs.org/oncorisk/>) is currently inaccessible. Effective utilization of any web-based analytical platform relies heavily on clear, comprehensive, and readily accessible documentation—especially when users are required to upload data in specific formats. The absence of functional tutorials or user manuals significantly limits usability and may hinder broader adoption of the resource.

We apologize for the broken links. As we replied to **Reviewer #1**, who also mentioned this: the documentation has been migrated and is now fully accessible at the main page of the OncoRisk when clicking the button “Documentation” and “Get Help” from “Help & Support”, which links to our correct document file <https://xiyasong.github.io/OncoRisk/>. This document provides the general information, the help pages for analysis modules, and a technical contact email.

2. Several figures included in the manuscript are of low resolution and remain difficult to interpret even when enlarged. I strongly recommend that the authors provide higher-quality images with adequate resolution to ensure readability and to allow proper technical evaluation.

We are very appreciative of this suggestion. We have now fixed this by improving the quality of all images, especially the screenshots used in the figures, to make sure everything is clear and readable. More importantly, we have split the old Figure 2 into two separate figures (the new **Figure 2**: Knowledge Base, Quick Search, and Network Tools; and the new **Figure 3**: Deep Query module), see our replies to **Reviewer #2**. This makes the figure layout much clearer, allows each picture to take up more space, and ensures that the readability is significantly improved.

We also manually checked that all other figures are clear and readable.

3. Although the manuscript describes the web server implementation and briefly mentions limitations, additional details regarding technical and computational constraints would strengthen the work. Specifically, the authors should clarify:

Thank you for your valuable suggestion. We have reframed the “Web server implementation” in the methodology section into three main aspects: “System architecture and user interface”, “Computational performance and scalability”, and “Data robustness and privacy” to clarify all the points the reviewer mentioned:

a) the maximum dataset size supported for upload, analysis, and visualization;

There are two parts of analysis related to file upload: server-maftools MAF file upload and OncoRisk pipeline analysis for cancer somatic mutation vcf files. The Server-Maftools module currently supports the upload and visualization of MAF files up to 100 MB. The OncoRisk VCF pipeline analysis is designed for the tumor tissue somatic sequencing data, which should only include somatic mutations and discard the germline mutations. While our standard demo files (generated by Illumina TruSight Oncology 500 (TSO500) High-Throughput Assay for somatic mutation calling) are approximately 6 MB for near-instantaneous processing, we support a maximum VCF upload size of 75 MB to ensure network stability and efficient Ensembl VEP-based annotation.

We have now emphasized their single upload data size limit in the section of “**Computational performance and scalability**”: “Users are recommended to upload somatic mutation profiles only, with a maximum size of 75 MB”(Lines 684-685), and for Server-Maftools: “For Server-Maftools, uploaded MAF files (≤100 MB) were processed into MAF objects through maftools with validation routines for required data columns.” (Lines 693-694)

We also make sure the upload size hint exists in the webpage, correspondingly in pipeline analysis and Server-Maftools:

Upload Genomic Files

Upload VCF, DNA, or RNA files for analysis. Supported formats include .vcf.gz, .json.gz and .tsv.

▲ Maximum file size: 75MB

Upload Your MAF File

Browse... Choose MAF file

Upload your MAF file

Supported formats: .maf, .txt, .tsv

Maximum file size: 100 MB

b) how the platform handles incorrectly formatted or invalid input data;

The platform is designed to be permissive at the ingestion stage to avoid disrupting user workflows. Even if uploaded files or search inputs do not fully match the expected formats, the system still accepts and processes them. However, depending on the error, incorrectly formatted inputs may result in incomplete analyses or no results. This design choice prioritizes service continuity and robustness, ensuring that malformed data does not crash the system or compromise downstream analyses.

We have updated this information in the methods section of the "**Data robustness and privacy**:" For the uploaded files or search inputs that do not fully conform to expected formats, the system still accepts and processes them to maintain system robustness. However, these incorrectly formatted or invalid inputs may lead to incomplete analyses or the absence of returned results, depending on the nature of the deviation."(Lines 701-704)

c) and the number of concurrent users or simultaneous uploads that the system can reasonably support.

Addressing these points would improve transparency, reproducibility, and user confidence in the robustness and scalability of the OncoRisk platform.

The Server-Maftools and Pan-Cancer-Visual is a Shiny app hosted on SciLifeLab Serve(<https://serve.scilifelab.se/>). We have added this into the manuscript under "**Computational performance and scalability**": "Server-Maftools and Pan-Cancer-Visual are hosted on the server that allocates 2vCPU and 4 GB of memory/RAM." (Lines 692-694). Currently, SciLifeLab Server **does not have a specific enforced limit** on the number of concurrent users, while they try to accommodate as many users as possible (<https://serve.scilifelab.se/docs/application-hosting/shiny/>).

For the pipeline analysis, it is built on a horizontally scalable architecture with a queue-based processing model for uploaded files. Incoming uploads are continuously accepted and placed into a unified processing queue, which is consumed in a controlled and uniform manner by parallel worker processes. This approach allows the system to absorb bursts of incoming data

without degradation of service and to process workloads efficiently in batches. At the current deployment stage, with 16 parallel workers and typical VCF file sizes of 4–6 MB, the system achieves a throughput of approximately 15,000 VCF files per day. Capacity can be increased linearly by adding additional worker instances that require minimal configuration and do not necessitate architectural changes.

We have added a description: "To ensure high concurrency, the system utilizes 16 parallel workers and SQL connection pooling, achieving a throughput of approximately 15,000 somatic VCF files (each around 4-6MB per file) per day. To ensure low-latency interactive performance, frequently accessed database partitions are indexed and resident in RAM for hot in-memory data serving, minimizing disk I/O. Furthermore, optimized SQL connection pooling enables high levels of concurrent access without contention, maintaining predictable performance even under increasing user load."(Lines 685 to 691)

For interactive platform usage, including browsing, querying, and result visualization, the system employs hot in-memory data serving combined with optimized SQL connection pooling. Frequently accessed database partitions are indexed and resident in available RAM across worker nodes, minimizing disk I/O and avoiding expensive database lookups. Efficient connection pool management further enables high levels of concurrent access without contention. This is also now indicated in the manuscript:" To ensure low-latency interactive performance, frequently accessed database partitions are indexed and resident in RAM for hot in-memory data serving, minimizing disk I/O. Furthermore, optimized SQL connection pooling enables high levels of concurrent access without contention, maintaining predictable performance even under increasing user load. "(Lines 687-691)

Together, these design choices allow the platform to support a large number of simultaneous users with consistently low latency, while maintaining predictable performance under increasing load.

Manuscript#: COMMSBIO-25-10600A

I was unable to type my comments in the "Comments for Author" box on the MTS site, so I'm putting them here instead:

I think the authors have done a commendable job in responding thoroughly to all the reviewer comments, and I fully support publishing the manuscript.